# Regularized Q-Learning

**Han-Dong Lim**
Electrical Engineering, KAIST
limaries30@kaist.ac.kr

**Donghwan Lee**
Electrical Engineering, KAIST
donghwan@kaist.ac.kr

## Abstract

Q-learning is widely used algorithm in reinforcement learning (RL) community. Under the lookup table setting, its convergence is well established. However, its behavior is known to be unstable with the linear function approximation case. This paper develops a new Q-learning algorithm, called RegQ, that converges when linear function approximation is used. We prove that simply adding an appropriate regularization term ensures convergence of the algorithm. Its stability is established using a recent analysis tool based on switching system models. Moreover, we experimentally show that RegQ converges in environments where Q-learning with linear function approximation was known to diverge. An error bound on the solution where the algorithm converges is also given.

## 1 Introduction

Recently, RL has shown great success in various fields. For instance, Mnih et al. [2015] achieved human level performance in several video games in the Atari benchmark [Bellemare et al., 2013]. Since then, researches on deep RL algorithms have shown significant progresses [Lan et al.]. Although great success has been achieved in practice, there is still gap between theory and the practical success. Especially when off-policy, function approximation, and bootstrapping are used together, the algorithm may show unstable behaviors. This phenomenon is called the deadly triad [Sutton and Barto, 2018]. Famous counter-examples are given in Baird [1995], Tsitsiklis and Van Roy [1997].

For policy evaluation, especially for temporal-difference (TD) learning algorithm, there has been several algorithms to resolve the deadly triad issue. Bradtke and Barto [1996] uses the least-square method to compute a solution of TD-learning, but it suffers from $O(h^2)$ time complexity, where $h$ is number of features. Maei [2011], Sutton et al. [2009] developed gradient descent based methods which minimize the mean square projected Bellman error. Ghiassian et al. [2020] added regularization term to TD Correction (TDC) algorithm, which uses a single time scale step-size. Lee et al. [2022] introduced several variants of the gradient TD (GTD) algorithm under control theoretic frameworks. Sutton et al. [2016] re-weights some states to match the on-policy distribution to stabilize the off-policy TD-learning. Bharadwaj Diddigi et al. [2020] uses $l_2$ regularization to propose a new convergent off-policy TD-learning algorithm. Mahadevan et al. [2014] studied regularization on the off-policy TD-learning through the lens of primal dual method.

First presented by Watkins and Dayan [1992], Q-learning also suffers from divergence issues under the deadly triad. While there are convergence results under the look-up table setting [Watkins and Dayan, 1992, Jaakkola et al., 1994, Borkar and Meyn, 2000, Lee and He, 2020], even with the simple linear function approximation, the convergence is only guaranteed under strong assumptions [Melo et al., 2008, Lee and He, 2020, Yang and Wang, 2019].

The main goal of this paper is to propose a practical Q-learning algorithm, called regularized Q-learning (RegQ), that guarantees convergence under linear function approximation. We prove its convergence using the ordinary differential equation (O.D.E) analysis framework in Borkar and Meyn [2000] together with the switching system approach developed in Lee and He [2020]. As in Lee

and He [2020], we construct upper and lower comparison systems, and prove its global asymptotic stability based on switching system theories. Compared to the standard Q-learning in Watkins and Dayan [1992], a difference lies in the additional $l_2$ regularization term, which makes the algorithm relevantly simple. Moreover, compared to the previous works in Carvalho et al. [2020], Maei et al. [2010], our algorithm is single time-scale, and hence, shows faster convergence rates experimentally. Our algorithm directly uses bootstrapping rather than circumventing the issue in the deadly triad. Therefore, it could give a new insight into training reinforcement learning algorithms with function approximation without using the so-called target network technique introduced in Mnih et al. [2015]. The main contributions of this paper are summarized as follows:

1. A new single time-scale Q-learning algorithm with linear function approximation is proposed.

2. We provide a theoretical analysis on the solution of the projected Bellman equation where a regularization term is included.

3. We prove the convergence of the proposed algorithm based on the O.D.E approach together with the switching system model in Lee and He [2020].

4. We experimentally show that our algorithm performs faster than other two time-scale Q-learning algorithms in Carvalho et al. [2020], Maei et al. [2010].

**Related works**:

Several works [Melo et al., 2008, Lee and He, 2020, Yang and Wang, 2019] have relied on strong assumptions to guarantee convergence of Q-learning under linear function approximation. Melo et al. [2008] adopts an assumption on relation between behavior policy and target policy to guarantee convergence, which is not practical in general. Lee and He [2020] assumes a similar assumption to that of Melo et al. [2008] to ensure the convergence with the so-called switching system approach. Yang and Wang [2019] considered a transition matrix that can be represented by the feature values, which restricts the class of Markov chain.

Motivated by the empirical success of the deep Q-learning in Mnih et al. [2015], recent works in Zhang et al. [2021], Carvalho et al. [2020], Agarwal et al., Chen et al. [2023] use the target network to circumvent the bootstrapping issue and guarantee convergence. Carvalho et al. [2020] designed a two time-scale learning method motivated by the target network method. Zhang et al. [2021] uses $l_2$ regularization with the target network, while a projection step is involved, which makes it difficult to implement practically. Moreover, it also relies on a two time-scale learning method. Chen et al. [2023] used target network and truncation method to address the divergence issue. Agarwal et al. additionally uses the so-called experience replay technique with the target network. Furthermore, the optimality is only guaranteed under a specific type of Markov chain. Even though, the target network update can guarantee stability, it often leads to slow convergence rate [Kim et al., 2019].

Maei et al. [2010] suggested the so-called Greedy-GQ (gradient Q-learning) algorithm, but due to non-convexity of the objective function, it could converge to a local optima. Lu et al. [2021] used linear programming approach [Manne, 1960] to design convergent Q-learning algorithm under deterministic control systems. Devraj and Meyn [2017] proposed a Q-learning algorithm that minimizes asymptotic variance. However, it requires the assumption that the number of changes of policy are finite, and involves matrix inversion at each iteration. Meyn [2023] introduced an optimistic training scheme with modified Gibbs policy for Q-learning with linear function approximation, which guarantees existence of a solution of the projected Bellman equation, but not the convergence. Geist et al. [2019], Xi et al. [2024] considered regularization on the policy which address a different scenario than the regularization in our work.

$l_2$ regularization has been actively explored in the RL literature. Farahm et al. [2016] proposed a regularized policy iteration algorithm that addresses a regularized policy evaluation problem, followed by a policy improvement step. The authors derived a performance error bound. Zhang et al. [2021] studied regularized projected Bellman equation and proves that inside a certain ball, the solution of the regularized projected Bellman equation exist and is unique. Manek and Kolter [2022] studied fixed points of off-policy TD-learning algorithm with regularization showing that the bias of the solution caused by the regularization can be large under certain scenario. Nonetheless, the regularization method has been widely used in practice Farebrother et al. [2018], Piché et al. [2021].

## 2 Preliminaries and Notations

### 2.1 Markov Decision Process

We consider an infinite horizon Markov Decision Process (MDP), which consists of a tuple $\mathcal{M} = (\mathcal{S}, \mathcal{A}, P, r, \gamma)$, where the state space $\mathcal{S}$ and action space $\mathcal{A}$ are finite sets, $P$ denotes the transition probability, $r : \mathcal{S} \times \mathcal{A} \times \mathcal{S} \to \mathbb{R}$ is the reward, and $\gamma \in (0, 1)$ is the discount factor. Given a stochastic policy $\pi : \mathcal{S} \to \mathcal{P}(\mathcal{A})$, where $\mathcal{P}(\mathcal{A})$ is the set of probability distributions over $\mathcal{A}$, agent at the current state $s_k$ selects an action $a_k \sim \pi(\cdot|s_k)$, then the agent's state changes to the next state $s_{k+1} \sim P(\cdot|s_k, a_k)$, and receives reward $r_{k+1} := r(s_k, a_k, s_{k+1})$. A deterministic policy is a special stochastic policy, which can be defined simply as a mapping $\pi : \mathcal{S} \to \mathcal{A}$.

The objective of MDP is to find a deterministic optimal policy, denoted by $\pi^*$, such that the cumulative discounted rewards over infinite time horizons is maximized, i.e., $\pi^* := \arg\max_\pi \mathbb{E}\left[\sum_{k=0}^\infty \gamma^k r_k \,\middle|\, \pi\right]$, where $(s_0, a_0, s_1, a_1, \ldots)$ is a state-action trajectory generated by the Markov chain under policy $\pi$, and $\mathbb{E}[\cdot|\pi]$ is an expectation conditioned on the policy $\pi$. The Q-function under policy $\pi$ is defined as $Q^\pi(s, a) = \mathbb{E}\left[\sum_{k=0}^\infty \gamma^k r_k \,\middle|\, s_0 = s, a_0 = a, \pi\right]$, $(s, a) \in \mathcal{S} \times \mathcal{A}$, and the optimal Q-function is defined as $Q^*(s, a) = Q^{\pi^*}(s, a)$ for all $(s, a) \in \mathcal{S} \times \mathcal{A}$. Once $Q^*$ is known, then an optimal policy can be retrieved by the greedy action, i.e., $\pi^*(s) = \arg\max_{a \in \mathcal{A}} Q^*(s, a)$. Throughout, we assume that the Markov chain is time homogeneous so that the MDP is well posed, which is standard in the literature. It is known that the optimal Q-function satisfies the so-called Bellman equation expressed as follows:

$$Q^*(s, a) = \mathbb{E}\left[r_{k+1} + \max_{a_{k+1} \in \mathcal{A}} \gamma Q^*(s_{k+1}, a_{k+1}) \,\middle|\, (s_k, a_k) = (s, a)\right] := \mathcal{T}Q^*(s, a), \qquad (1)$$

where $\mathcal{T}$ is called the Bellman operator.

### 2.2 Notations

In this paper, we will use an O.D.E. model [Borkar and Meyn, 2000] of Q-learning to analyze its convergence. To this end, it is useful to introduce some notations in order to simplify the overall expressions. Throughout the paper, $e_a$ and $e_s$ denote $a$-th and $s$-th canonical basis vectors in $\mathbb{R}^{|\mathcal{A}|}$ and $\mathbb{R}^{|\mathcal{S}|}$, respectively, and $\otimes$ stands for the Kronecker product. Let us introduce the following notations:

$$P := \begin{bmatrix} P_1 \\ \vdots \\ P_{|\mathcal{A}|} \end{bmatrix} \in \mathbb{R}^{|\mathcal{S}||\mathcal{A}| \times |\mathcal{S}|}, \quad R := \begin{bmatrix} R_1 \\ \vdots \\ R_{|\mathcal{A}|} \end{bmatrix} \in \mathbb{R}^{|\mathcal{S}||\mathcal{A}|}, \quad Q := \begin{bmatrix} Q_1 \\ \vdots \\ Q_{|\mathcal{A}|} \end{bmatrix} \in \mathbb{R}^{|\mathcal{S}||\mathcal{A}|},$$

$$D_a := \begin{bmatrix} d(1, a) & & \\ & \ddots & \\ & & d(|\mathcal{S}|, a) \end{bmatrix} \in \mathbb{R}^{|\mathcal{S}| \times |\mathcal{S}|}, \quad D := \begin{bmatrix} D_1 & & \\ & \ddots & \\ & & D_{|\mathcal{A}|} \end{bmatrix} \in \mathbb{R}^{|\mathcal{S}||\mathcal{A}| \times |\mathcal{S}||\mathcal{A}|},$$

where $P_a \in \mathbb{R}^{|\mathcal{S}| \times |\mathcal{S}|}, a \in \mathcal{A}$ is the state transition matrix whose $i$-th row and $j$-th column component denotes the probability of transition to state $j$ when action $a$ is taken at state $i$, $P^\pi \in \mathbb{R}^{|\mathcal{S}||\mathcal{A}| \times |\mathcal{S}||\mathcal{A}|}$ represents the state-action transition matrix under policy $\pi$, i.e.,

$$(e_s \otimes e_a)^T P^\pi (e_{s'} \otimes e_{a'}) = \mathbb{P}[s_{k+1} = s', a_{k+1} = a' | s_k = s, a_k = a, \pi],$$

$Q_a = Q(\cdot, a) \in \mathbb{R}^{|\mathcal{S}|}, a \in \mathcal{A}$ and $R_a(s) := \mathbb{E}[r(s, a, s')|s, a], s \in \mathcal{S}$. Moreover, $d(\cdot, \cdot)$ is the state-action visit distribution, where i.i.d. random variables $\{(s_k, a_k)\}_{k=0}^\infty$ are sampled, i.e., $d(s, a) = \mathbb{P}[s_k = s, a_k = a]$, $(s, a) \in \mathcal{S} \times \mathcal{A}$. With a slight abuse of notation, $d$ will be also used to denote the vector $d \in \mathbb{R}^{|\mathcal{S}||\mathcal{A}|}$ such that $d^T(e_s \otimes e_a) = d(s, a)$, $\forall (s, a) \in \mathcal{S} \times \mathcal{A}$. In this paper, we represent a policy in a matrix form in order to formulate a switching system model. In particular, for a given policy $\pi$, define the matrix $\Pi^\pi \in \mathbb{R}^{|\mathcal{S}| \times |\mathcal{S}||\mathcal{A}|}$:

$$\Pi^\pi := \begin{bmatrix} (e_{\pi(1)} \otimes e_1) & (e_{\pi(2)} \otimes e_2) & \cdots & (e_{\pi(|\mathcal{S}|)} \otimes e_{|\mathcal{S}|}) \end{bmatrix}^\top.$$

Then, we can prove that for any deterministic policy, $\pi$, we have $\Pi^\pi Q = [Q(1, \pi(1)) \quad Q(2, \pi(2)) \quad \cdots \quad Q(|\mathcal{S}|, \pi(|\mathcal{S}|))]^T$. For simplicity, let $\Pi_Q := \Pi^\pi$ when $\pi(s) = \arg\max_{a \in \mathcal{A}} Q(s, a)$. Moreover, we can prove that for any deterministic policy $\pi$, $P^\pi = P\Pi^\pi \in \mathbb{R}^{|\mathcal{S}||\mathcal{A}| \times |\mathcal{S}||\mathcal{A}|}$, where $P^\pi$ is the state-action transition probability

matrix. Using the notations introduced, the Bellman equation in (1) can be compactly written as $Q^* = \gamma P \Pi_{Q^*} Q^* + R =: \mathcal{T} Q^*$, where $\pi_{Q^*}$ is the greedy policy defined as $\pi_{Q^*}(s) = \arg \max_{a \in \mathcal{A}} Q^*(s, a)$.

## 2.3 Q-learning with linear function approximation

Q-learning is widely used model-free learning to find $Q^*$, whose updates are given as

$$Q_{k+1}(s_k, a_k) \leftarrow Q_k(s_k, a_k) + \alpha_k \delta_k, \tag{2}$$

where $\delta_k = r_{k+1} + \gamma \max_{a \in \mathcal{A}} Q_k(s_{k+1}, a) - Q_k(s_k, a_k)$ is called the TD error. Each update uses an i.i.d. sample $(s_k, a_k, r_{k+1}, s_{k+1})$, where $(s_k, a_k)$ is sampled from a state-action distribution $d(\cdot, \cdot)$.

Here, we assume that the step-size is chosen to satisfy the so-called the Robbins-Monro condition [Robbins and Monro, 1951], $\alpha_k > 0$, $\sum_{k=0}^{\infty} \alpha_k = \infty$, $\sum_{k=0}^{\infty} \alpha_k^2 < \infty$. When the state-spaces and action-spaces are too large, then the memory and computational complexities usually become intractable. In such a case, function approximation is commonly used to approximate Q-function [Mnih et al., 2015, Hessel et al., 2018]. Linear function approximation is one of the simplest function approximation approaches. In particular, we use the feature matrix $X \in \mathbb{R}^{|\mathcal{S}||\mathcal{A}| \times h}$ and parameter vector $\theta \in \mathbb{R}^h$ to approximate Q-function, i.e., $Q \simeq X\theta$, where the feature matrix is expressed as $X := [x(1, 1) \quad \cdots \quad x(1, |\mathcal{A}|) \quad \cdots \quad x(|\mathcal{S}|, |\mathcal{A}|)]^T \in \mathbb{R}^{|\mathcal{S}||\mathcal{A}| \times h}$. Here, $x(\cdot, \cdot) \in \mathbb{R}^h$ is called the feature vector, and $h$ is a positive integer with $h \ll |\mathcal{S}||\mathcal{A}|$. The corresponding greedy policy becomes $\pi_{X\theta}(s) = \arg \max_{a \in \mathcal{A}} x(s, a)^T \theta$. Note that the number of policies characterized by the greedy policy is finite. This is because the policy is invariant under constant multiplications, and there exists a finite number of sectors on which the policy is invariant. Next, we summarize some standard assumptions adapted throughout this paper.

**Assumption 2.1.** *The state-action visit distribution is positive, i.e., $d(s, a) > 0$ for all $(s, a) \in \mathcal{S} \times \mathcal{A}$.*

**Assumption 2.2.** *The feature matrix, $X$, has full column rank, and is a non-negative matrix. Moreover, columns of $X$ are orthogonal.*

**Assumption 2.3** (Boundedness on feature matrix and reward matrix). *There exists constants, $X_{\max} > 0$ and $R_{\max} > 0$, such that $\max(||X||_\infty, ||X^T||_\infty) < X_{\max}$ and $||R||_\infty < R_{\max}$.*

We note that except for the orthogonality of the feature matrix in Assumption 2.2, the assumptions in the above are commonly adopted in the literature, e.g. Carvalho et al. [2020], Lee and He [2020]. Moreover, under Assumption 2.1, $D$ is a nonsingular matrix with strictly positive diagonal elements.

**Lemma 2.4** (Gosavi [2006]). *Under Assumption 2.3, $Q^*$, is bounded, i.e., $||Q^*||_\infty \leq \frac{R_{\max}}{1-\gamma}$.*

The proof of Lemma 2.4 comes from the fact that under the discounted infinite horizon setting, $Q^*$ can be expressed as an infinite sum of a geometric sequence.

## 2.4 Switching System

In this paper, we consider a particular system, called the *switched linear system* [Liberzon, 2003],

$$\dot{x}_t = A_{\sigma_t} x_t, \quad x_0 = z \in \mathbb{R}^n, \quad t \in \mathbb{R}_+, \tag{3}$$

where $x_t \in \mathbb{R}^n$ is the state, $\mathcal{M} := \{1, 2, \ldots, M\}$ is called the set of modes, $\sigma_t \in \mathcal{M}$ is called the switching signal, and $\{A_\sigma, \sigma \in \mathcal{M}\}$ are called the subsystem matrices. The switching signal can be either arbitrary or controlled by the user under a certain switching policy.

Stability and stabilization of (3) have been widely studied for decades. Still, finding a practical and effective condition for them is known to be a challenging open problem. Contrary to linear time-invariant systems, even if each subsystem matrix $A_\sigma$ is Hurwitz, the overall switching system may not be stable in general. This tells us that tools in linear system theories cannot be directly applied to conclude the stability of the switching system.

Another approach is to use the Lyapunov theory [Khalil, 2002]. From standard results in control system theories, finding a Lyapunov function ensures stability of the switching system. If the switching system consists of matrices with strictly negatively row dominant diagonals, defined in Definiiton A.5 in the Appendix, or negative-definite matrices, we can always find a common (piecewise) quadratic Lyapunov function to ensure its stability. We use this fact to prove the convergence of the proposed algorithm.

**Lemma 2.5.** *Consider a switched system in (3). Suppose one of the following two conditions hold:*

1) *Each $A_\sigma$ for $\sigma \in \mathcal{M}$ has a strictly negatively row dominating diagonal, i.e., $[A_\sigma]_{ii} + \sum_{j \in \{1,2,...,n\} \setminus \{i\}} |[A_\sigma]_{ij}| < 0$ for all $1 \le i \le n$.*
2) *$A_\sigma + A_\sigma^\top \prec 0$ for all $\sigma \in \mathcal{M}$.*

*Then, the origin of (3) is asymptotically stable.*

The proof is given in Appendix A.4

## 3 Projected Bellman equation

In this section, we introduce the notion of projected Bellman equation with a regularization term, and establish connections between it and the proposed algorithm. Moreover, we briefly discuss the existence and uniqueness of the solution of the projected Bellman equation. We will also provide an example to illustrate the existence and uniqueness of the solution.

### 3.1 Projected Bellman equation (PBE)

When using the linear function approximation, since the true action value may not lie in the subspace spanned by the feature vectors, a solution of the Bellman equation may not exist in general. To resolve this issue, a standard approach is to consider the projected Bellman equation (PBE) defined as

$$X\theta^* = \Gamma \mathcal{T} X\theta^*, \tag{4}$$

where $\Gamma := X(X^T D X)^{-1} X^T D$ is the weighted Euclidean projection with respect to state-action visit distribution onto the subspace spanned by the feature vectors, and $\mathcal{T} X\theta^* = \gamma P \Pi_{X\theta^*} X\theta^* + R$. In this case, there are more chances for a solution satisfying the PBE to exist. Still, there may exist cases where the PBE does not admit a solution. To proceed, letting

$$A_{\pi_{X\theta^*}} := X^T D X - \gamma X^T D P \Pi_{X\theta^*} X, \quad b = X^T D R,$$

we can rewrite (4) equivalently as

$$X\theta^* = X(X^T D X)^{-1} X^T D(\gamma P \Pi_{X\theta^*} X\theta^* + R) \Leftrightarrow A_{\pi_{X\theta^*}} \theta^* = b, \tag{5}$$

Furthermore, we use the simplified notation $C := X^T D X$. A potential deterministic algorithm to solve the above equation is

$$\theta_{k+1} = \theta_k + \alpha_k(b - A_{\pi_{X\theta_k}} \theta_k). \tag{6}$$

It iteratively solves the linear or nonlinear equation, which is a widely used algorithm called a Richardson iteration [Kelley, 1995]. If it converges, i.e., $\theta_k \to \theta^*$ as $k \to \infty$, then it is clear that $\theta^*$ solves (5). In this paper, the proposed algorithm is a stochastic algorithm that solves the modified equation

$$b - (A_{\pi_{X\theta_\eta^*}} + \eta I)\theta_\eta^* = 0, \tag{7}$$

where $I$ is the $h \times h$ identity matrix, and $\eta \ge 0$ is a weight on the regularization term. We can use $\eta C$ instead of $\eta I$ as the regularization term but $\eta C$ is known to solve a MDP with modified discount factor Chen et al. [2023]. Similar to (6), the corresponding deterministic algorithm is

$$\theta_{k+1} = \theta_k + \alpha_k(b - (A_{\pi_{X\theta_k}} + \eta I)\theta_k). \tag{8}$$

If it converges, i.e., $\theta_k \to \theta_\eta^*$ as $k \to \infty$, then it is clear that $\theta_\eta^*$ solves (7).

### 3.2 Regularized projected Bellman equation

The equation (7) can be written as the regularized projected Bellman equation (RPBE)

$$X\theta_\eta^* = \Gamma_\eta \mathcal{T} X\theta_\eta^*, \tag{9}$$

where

$$\Gamma_\eta := X(X^\top D X + \eta I)^{-1} X^\top D. \tag{10}$$

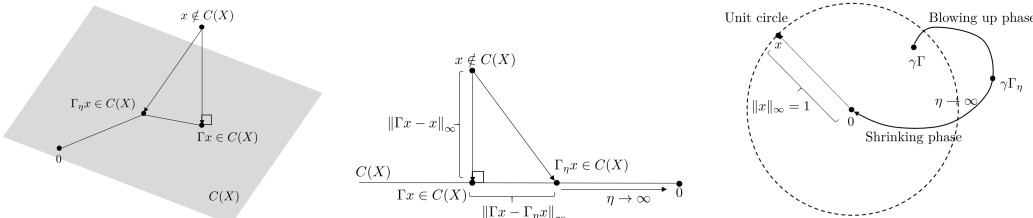

(a) Regularized projection: $C(X)$ means the range space of $X$

(b) Regularized projection: One dimensional case

(c) Boundedness of the projection

Figure 1: Illustrative explanation on the regularized projection. The Figure 1c implies that as $\eta \to \infty$, $\gamma\Gamma_\eta$ can potentially move outside of the unit ball satisfying $||x||_\infty \leq 1$, and this phase is indicated with the term "blowing up" phase. The quantity $||\gamma\Gamma_\eta||_\infty$ actually blows up initially as $\eta \to \infty$. However, since $\lim_{\eta\to\infty}||\gamma\Gamma_\eta||_\infty = 0$, we know that $\gamma\Gamma_\eta$ will eventually converge to the origin and move inside the unit ball. This behavior is indicated by the "shrinking" phase in the figure.

The proof of the equivalence between (7) and (9) are given in Lemma A.12 in the Appendix Section A.3. The matrix $\Gamma_\eta$ can be viewed as a modified projection operator which will be called the regularized projection. It can be interpreted as the projection with a regularization term $\Gamma_\eta(x) = \arg\min_{\theta\in\mathbb{R}^h} \left( \frac{1}{2} \|x - X\theta\|_D^2 + \frac{\eta}{2} \|\theta\|_2^2 \right)$. The concept is illustrated in Figure 1a. Before moving forward, some natural questions that arise here are as follows: How does $\theta^*$ and $\theta_\eta^*$ differ? Furthermore, which conditions can determine the existence and uniqueness of the solution of (4) and (9)? Partial answers are given in the sequel.

First, let us assume that the solution of (4) and (9), $\theta^*$ and $\theta_\eta^*$, respectively, exist and are unique. To understand the difference between $\theta^*$ and $\theta_\eta^*$, an important property of $\Gamma_\eta$ is introduced:

**Lemma 3.1.** *(a) The projection $\Gamma_\eta$ satisfies the following properties:* $\lim_{\eta\to\infty} \Gamma_\eta = 0$ *and* $\lim_{\eta\to 0} \Gamma_\eta = \Gamma$.

*(b) We have* $\|\Gamma_\eta\|_\infty \leq \left\|X^\top D\right\|_2 \|X\|_2 \left\|(X^\top DX)^{-1}\right\|_2 \sqrt{|S \times A|}$ *for all $\eta \geq 0$.*

The proof is given in Appendix A.5. From the above result, one can observe that as $\eta \to \infty$, the projection is attracted to the origin as illustrated in Figure 1b. Moreover, as $\eta \to 0$, we will expect that $\theta_\eta^* \to \theta^*$. Furthermore, one can observe that the bound in item (b) of Lemma 3.1 cannot be controlled by simply scaling the feature function, and therefore, it more depends on the inherent structures of the feature matrix $X$. The concept is illustrated in Figure 1c. We will provide a more in-depth discussion on the error bound of $\theta_\eta^* - \theta^*$ in Section 3.3.

Now, we will discuss the existence and uniqueness of the solutions. Considering the non-existence of the solution of (4) [De Farias and Van Roy, 2000], (9) may not also have a solution. However, for RPBE in (9), we can prove that under mild conditions, its solution exists and is unique. We provided an example where the solution does not exist for (4) but does exist for (9) in Appendix A.14. Let us first state a general condition such that the solution of (9) exists and is unique:

$$\gamma||\Gamma_\eta||_\infty < 1, \tag{11}$$

**Lemma 3.2.** *Suppose that (11) holds. Then the solution of RPBE in (9) exists and is unique.*

The proof is given in Appendix A.6, which uses Banach fixed-point theorem [Agarwal et al., 2018]. From Lemma 3.2, we can see that the condition, $\gamma||\Gamma_\eta||_\infty < 1$, is important to guarantee the uniqueness and existence of the solution. We will clarify under what situations the condition (11) can be met, and provide related discussions in Lemma 3.3, 3.4, and 3.5, where each lemma illustrate different scenarios when (11) is met. In particular, Lemma 3.3 shows that with simple feature scaling, $\eta$ can be easily chosen such that (11) holds. Furthermore, Lemma 3.4 considers a case when $\eta$ is in a small neighborhood of zero, and Lemma 3.5 considers the case when (11) should hold for all $\eta \geq 0$.

We note that Zhang et al. [2021] also studied the solution of the regularized projected Bellman equation. Nonetheless, the result of Zhang et al. [2021] only ensures a unique solution within a certain ball whereas we consider the whole $\mathbb{R}^h$ space.

**Lemma 3.3.** *For $\eta > \gamma||X^\top D||_\infty||X||_\infty + ||X^\top DX||_\infty$, we have $\gamma\|\Gamma_\eta\|_\infty < 1$.*

The proof is given in Lemma A.11 in Appendix. From Lemma 3.3, we can satisfy the condition in (11) with scaling the values of the feature matrix $X$. For example, if $\max(||X||_\infty, ||X^\top||_\infty) < 1$, it is enough to choose $\eta > 2$ to meet the condition in Lemma 3.3. It is worth noting that scaling the values of feature matrix is a commonly employed technique in the both theoretical literature or in practice.

**Lemma 3.4.** *Suppose $\gamma||\Gamma||_\infty < 1$ so that the solution point of PBE in (4) exists and is unique. Then, the condition $0 \leq \eta < \frac{(1-\gamma||\Gamma||_\infty)||(X^T DX)^{-1}||_\infty^{-1}}{\gamma||(X^T DX)^{-1}||_\infty||X||_\infty||X^T D||_\infty + (1-\gamma||\Gamma||_\infty)}$ implies $\gamma||\Gamma_\eta||_\infty < 1$.*

The proof is in Appendix A.7. We note that the condition, $\gamma||\Gamma||_\infty < 1$ in Lemma 3.4 is used to guarantee the existence and uniqueness of the solution of PBE in (4), which is provided in Melo et al. [2008]. Therefore, without any special conditions, we can guarantee the existence and uniqueness of the solution in the neighborhood of $\eta = 0$.

**Lemma 3.5.** *Suppose the feature vector satisfies $X^\top DX = aI$ for a positive real number $a$ such that $a|S||A| \geq 1$. Assume that $||X||_2 \leq 1$ and $D = 1/(|\mathcal{S}||\mathcal{A}|)I$. Then, (11) holds for all $\eta > 0$.*

The proof is given in Appendix A.8. A simple example where the above statement holds is by letting $X = I$. This is only a conceptual example and there could be many other examples in existence.

### 3.3 Error analysis

As promised in the previous section, we provide discussion on the behavior and quality of $\theta_\eta^*$, i.e., the error bound analysis in $\theta^* - \theta_\eta^*$ depending on $\eta$. Even though Manek and Kolter [2022] provided a specific example when the bias can be large in the policy evaluation case, throughout the analysis, we show that the error can be small under particular scenarios.

Let us first examine the case when $\eta \to 0$ and $\eta \to \infty$. As discussed in Section 3.2, we can consider $\eta \to 0$ if we can guarantee the existence of $\theta_\eta^*$ and $\theta^*$ when $\eta$ is nearby the origin, for example in the case of Lemma 3.4 and 3.5. As $\eta \to 0$, (4) and (9) coincide, implying that $\theta_\eta^* \to \theta^*$.

Furthermore, as $\eta$ gets larger, by Lemma 3.3, we can always guarantee existence and uniqueness of $\theta_\eta^*$ after a certain threshold. As from the discussion of Lemma 3.1, we expect $\theta_\eta^* \to 0$, which is stated in the following lemma whose proof is given in Appendix A.9:

**Lemma 3.6.** *We have $\lim_{\eta \to \infty} \theta_\eta^* = 0$.*

Note that even if a solution satisfying (7) exists, $X\theta_\eta^*$ may be different from $Q^*$. However, we can derive a bound on the error, $X\theta_\eta^* - Q^*$, using simple algebraic inequalities and contraction property of the Bellman operator. We present the error bound of the solution in the following lemma:

**Lemma 3.7.** *Suppose (11) holds. Then, we have :$||X\theta_\eta^* - Q^*||_\infty \leq \frac{1}{1-\gamma||\Gamma_\eta||_\infty}||\Gamma_\eta Q^* - Q^*||_\infty$.*

The proof is given in Appendix A.10. We provide a discussion on the error bound in the following:

1) $\eta \to 0$: Consider the case when $\theta_\eta^*$ and $\theta^*$ exists and unique, for example the condition in Lemma 3.4 is satisfied. Since $\Gamma_\eta \to \Gamma$ from Lemma 3.1, we exactly recover the error bound by fixed point of original projected Bellman equation ($\eta = 0$) in (4), which is $\frac{||\Gamma Q^* - Q^*||_\infty}{1-\gamma||\Gamma||_\infty}$ provided in Melo et al. [2008]. Thus, our bound in Lemma 3.7 is tight when $\eta \to 0$.

2) $\eta \to \infty$: As from Lemma 3.3, $\theta_\eta^*$ always exist when $\eta$ gets larger than certain value. Noting that $\Gamma_\eta \to 0$, we have $||X\theta_\eta^* - Q^*||_\infty \leq ||Q^*||_\infty$. Considering that $\theta_\eta^* \to 0$ as $\eta \to \infty$ from Lemma 3.6, we should have $||X \cdot 0 - Q^*||_\infty = ||Q^*||_\infty$. Thus, our bound in Lemma 3.7 is tight when $\eta \to \infty$.

3) The error bound is close to zero: An upper bound on Lemma 3.7 can be obtained by simple algebraic manipulation:

$$||X\theta_\eta^* - Q^*||_\infty \leq \frac{||\Gamma_\eta Q^* - Q^*||_\infty}{1-\gamma||\Gamma_\eta||_\infty} \leq \frac{1}{1-\gamma||\Gamma_\eta||_\infty}\left(\underbrace{||\Gamma_\eta Q^* - \Gamma Q^*||_\infty}_{\text{(T1)}} + \underbrace{||\Gamma Q^* - Q^*||_\infty}_{\text{(T2)}}\right).$$
(12)

Suppose that the features are well designed such that (T2) in (12) will be small. For example, if $Q^*$ is in the range space of $X$, then the error term in (T2) vanishes. Moreover, we can make (T1) arbitrarily

small as follows: as $\eta \to 0$, we have $||\Gamma_\eta - \Gamma||_\infty \to 0$ while $1 - \gamma||\Gamma_\eta||_\infty > 0$. This yields (T1) in (12) to be sufficiently small. In the end, we will have $||X\theta_\eta^* - Q^*||_\infty \le \epsilon$ for any $\epsilon \ge 0$.

4) When the PBE does not admit a fixed point around $\eta = 0$: In this case, we should always choose $\eta > 0$ greater than a certain number, and (T1) cannot be entirely vanished, while (T2) can be arbitrarily close to zero when $Q^*$ is close to the range space of $X$. The error in (T1) cannot be overcame because it can be seen as a fundamental error caused by the regularization for PBE. However, (T1) can be still small enough in many cases when $||\Gamma - \Gamma_\eta||_\infty$ is small.

## 4 Algorithm

In this section, we will introduce our main algorithm, called RegQ, and elaborate the condition on the regularization term to make the algorithm convergent. The proposed algorithm is motivated by TD-learning. In particular, for on-policy TD-learning, one can establish its convergence using the property of the stationary distribution. On the other hand, for an off-policy case, the mismatch between the sampling distribution and the stationary distribution could cause its divergence [Sutton et al., 2016]. To address this problem, Bharadwaj Diddigi et al. [2020] adds a regularization term to TD-learning in order to make it convergent. Since Q-learning can be interpreted as an off-policy TD-learning, we add a regularization term to Q-learning update motivated by Bharadwaj Diddigi et al. [2020]. This modification leads to the proposed RegQ algorithm as follows:

$$\theta_{k+1} = \theta_k + \alpha_k(x(s_k, a_k)\delta_k - \eta\theta_k) \tag{13}$$

The pseudo-code is given in Appendix A.16. Note that it can be viewed as a gradient descent step applied to the TD-loss $L(\theta) := \frac{1}{2}(y_k - Q_\theta(s_k, a_k))^2 + \frac{1}{2}\eta \|\theta\|_2^2$, where $y_k = r_{k+1} + \gamma\max_{a \in A}Q_{\theta_k}(s_{k+1}, a)$ is the TD-target, and $Q_{\theta_k} = X\theta_k$. Furthermore, letting $\eta = 0$, the above update is reduced to the standard Q-learning with linear function approximation in (2). The proposed RegQ is different from Bharadwaj Diddigi et al. [2020] in the sense that a regularization term is applied to Q-learning instead of TD-learning. Rewriting the stochastic update in a deterministic manner, it can be written as follows:

$$\theta_{k+1} = \theta_k + \alpha_k(b - (A_{\pi_{X\theta_k}} + \eta I)\theta_k + m_{k+1}), \tag{14}$$

where $m_{k+1} = \delta_k x(s_k, a_k) - \eta\theta_k - (b - (A_{\pi_{X\theta_k}} + \eta I)\theta_k)$ is a Martingale difference sequence. Without $m_{k+1}$, (14) is reduced to the deterministic version in (8). In our convergence analysis, we will apply the O.D.E. approach, and in this case, $A_{\pi_{X\theta_k}} + \eta I$ will determine the stability of the corresponding O.D.E. model, and hence, convergence of (13). Note that (14) can be interpreted as a switching system defined in (3) with stochastic noises. As mentioned earlier, proving the stability of a general switching system is challenging in general. However, we can find a common Lyapunov function to prove its asymptotic stability. In particular, we can make $-(A_{\pi_{X\theta_k}} + \eta I)$ to have a strictly negatively row dominant diagonal or negative-definite under the following condition:

$$\eta > \min\left\{\underbrace{\gamma||X^\top D||_\infty||X||_\infty + ||X^\top DX||_\infty}_{(S1)}, \underbrace{\lambda_{\max}(C)\left(\max_{\substack{\pi \in \Theta \\ (s,a) \in \mathcal{S}\times\mathcal{A}}} \frac{\gamma d^T P^\pi(e_a \otimes e_s)}{2d(s,a)} - \frac{2-\gamma}{2}\right)}_{(S2)}\right\}, \tag{15}$$

The conditions in $(S1)$ and $(S2)$, which make $-(A_{\pi_{X\theta_k}} + \eta I)$ to have strictly negatively row dominant diagonal or negative definite matrix , respectively, do not necessarily imply each others, which are discussed in Appendix A.15. Now, we can use the Lyapunov argument to establish stability of the overall system. Building on the fact, in the next section, we prove that under the stochastic update (13), we have $\theta_k \to \theta_\eta^*$ as $k \to \infty$ with probability one, where $\theta_\eta^*$ satisfies RPBE in (7). If $\eta = 0$ satisfies (15), we can guarantee convergence to an optimal policy without errors.

## 5 Convergence Analysis

Recently, Lee and He [2020] suggested a switching system framework to prove the stability of Q-learning in the linear function approximation cases. However, its assumption on the behavior

policy and feature matrix seems too stringent to check in practice. Here, we develop more practical Q-learning algorithm by adding an appropriately preconditioned regularization term. We prove the convergence of the proposed Q-learning with regularization term (13) following lines similar to Lee and He [2020]. Our proof mainly relies on Borkar-Meyn theorem. Therefore, we first discuss about the corresponding O.D.E. for the proposed update in (13), which is

$$\dot{\theta}_t = -(X^T D X + \eta I)\theta_t + \gamma X^T D P \Pi_{X\theta_t} X \theta_t + X^T D R := f(\theta_t). \tag{16}$$

Then, using changes of coordinates, the above O.D.E. can be rewritten as

$$\frac{d}{dt}(\theta_t - \theta_\eta^*) = (-A_{\pi_{X\theta_t}} - \eta I)(\theta_t - \theta_\eta^*) + \gamma X^T D P (\Pi_{X\theta_t} - \Pi_{X\theta_\eta^*}) X \theta_\eta^*, \tag{17}$$

where $\theta_\eta^*$ satisfies (7). Here, we assume that an equilibrium point exists and is unique. We later prove that if an equilibrium exists, then it is unique. To apply Borkar-Meyn theorem in Lemma A.1, we discuss about the asymptotic stability of the O.D.E. in (17). Note that (17) includes an affine term, i.e., it cannot be expressed as a matrix times vector $\theta_t - \theta_\eta^*$. It is in general hard to establish asymptotic stability of switched linear system with affine term compared to switched linear system (3). To circumvent this difficulty, Lee and He [2020] proposed upper and lower comparison systems, which upper bounds and lower bounds the original system. Then, the stability of the original system can be established by proving the stability of the upper and lower systems, which are easier to analyze. Following similar lines, to check global asymptotic stability of the original system, we also introduce upper and lower comparison systems. Then, we prove global asymptotic stability of the two bounding systems. Since upper and lower comparison systems can be viewed as switched linear system and linear system, respectively, the global asymptotic stability is easier to prove. We stress that although the switching system approach in Lee and He [2020] is applied in this paper, the detailed proof is entirely different and nontrivial. In particular, the upper and lower comparison systems are given as follows:

$$\dot{\theta}_t^u = (-X^T D X - \eta I + \gamma X^T D P \Pi_{X\theta_t^u} X)\theta_t^u, \quad \dot{\theta}_t^l = (-X^T D X - \eta I + \gamma X^T D P \Pi_{X\theta_\eta^*} X)\theta_t^l,$$

where $\theta_t^u$ and $\theta_t^l$ denote the states of the upper and lower systems, respectively. We defer the detailed construction of each system to Appendix A.12. The stability of overall system can be proved by establishing stability of the upper and lower comparison systems.

**Theorem 5.1.** *Suppose $\eta$ satisfies (15), and Assumption 2.1, 2.2, and 2.3 hold. Moreover, assume that a solution of RPBE in (7) exists. Then, it is also unique, and the origin is the unique globally asymptotically stable equilibrium point of (17).*

The detailed proof is given in Appendix A.12. Building on the previous results, we now use Borkar and Meyn's theorem in Lemma A.1 to establish the convergence of RegQ. The full proof of the following theorem is given in Appendix A.13.

**Theorem 5.2.** *Suppose $\eta$ satisfies (15), then with Assumption 2.1, 2.2, and 2.3 holds. Assume that solution of RPBE in (7) exists. Then, $\theta_\eta^*$ is unique, and under the stochastic update (13), $\theta_k \to \theta_\eta^*$ as $k \to \infty$ with probability one, where $\theta_\eta^*$ satisfies (7).*

We note that if $\eta$ is larger than the term $(S_1)$ in (15), then $\theta_\eta^*$ exists and is unique by Lemma 3.3.

## 6 Experiments

In this section, we briefly present the experimental results under well-known environments in Tsitsiklis and Van Roy [1996], Baird [1995], where Q-learning with linear function approximation diverges. As from Figure 2b, our algorithm shows faster convergence rate than other algorithms. Further details on the experiments are deferred to Appendix B. In Appendix B.6, we also compare performance under the Mountain Car environment [Sutton and Barto, 2018] where Q-learning performs well. In Appendix B.5, we show experimental results under various step-size and $\eta$. Moreover, the trajectories of upper and lower systems to illustrate the theoretical results are given in Appendix B.7.

## 7 Conclusion

In this paper, we presented a new convergent Q-learning with linear function approximation (RegQ), which is simple to implement. We provided theoretical analysis on the proposed RegQ, and demonstrated its performance on several experiments, where the original Q-learning with linear function

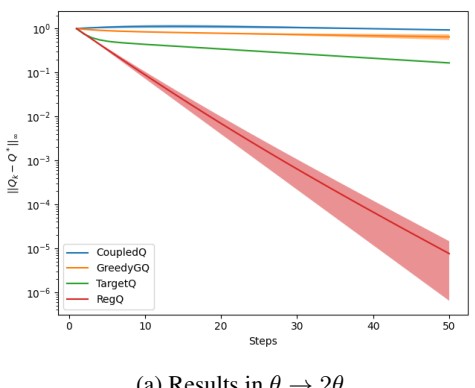 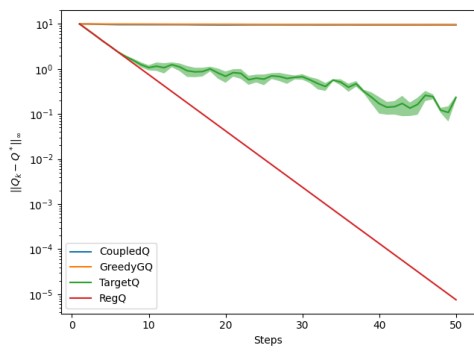

(a) Results in $\theta \to 2\theta$

(b) Results in Baird seven star counter example

Figure 2: Experiment results

approximation diverges. Developing a new Q-learning algorithm with linear function approximation without bias would be one interesting future research topic. Moreover, considering the great success of deep learning, it would be interesting to develop deep reinforcement learning algorithms with appropriately preconditioned regularization term instead of using the target network.

# 8 Acknowledgements

The work was supported by the Institute of Information Communications Technology Planning Evaluation (IITP) funded by the Korea government under Grant 2022-0-00469, and the BK21 FOUR from the Ministry of Education (Republic of Korea).

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

# A  Appendix

## A.1  O.D.E analysis

The dynamic system framework has been widely used to prove convergence of reinforcement learning algorithms, e.g., Sutton et al. [2009], Maei et al. [2010], Borkar and Meyn [2000], Lee and He [2020]. Especially, Borkar and Meyn [2000] is one of the most widely used techniques to prove stability of stochastic approximation using O.D.E. analysis. Consider the following stochastic algorithm with a nonlinear mapping $f : \mathbb{R}^n \to \mathbb{R}^n$:

$$\theta_{k+1} = f(\theta_k) + m_k, \tag{18}$$

where $m_k \in \mathbb{R}^n$ is an i.i.d. noise vector. For completeness, results in Borkar and Meyn [2000] are briefly reviewed in the sequel. Under Assumption A.2 given in Appendix A.2, we now introduce Borkar and Meyn theorem below.

**Lemma A.1** (Borkar and Meyn theorem). *Suppose that Assumption A.2 in the  Appendix A.2 holds, and consider the stochastic algorithm in (18). Then, for any initial $\theta_0 \in \mathbb{R}^n$, $\sup_{k \geq 0} ||\theta_k|| < \infty$ with probability one. In addition , $\theta_k \to \theta^e$ as $k \to \infty$ with probability one, where $\theta^e$ satisfies $f(\theta^e) = 0$.*

The main idea of Borkar and Meyn theorem is as follows: iterations of a stochastic recursive algorithm follow the solution of its corresponding O.D.E. in the limit when the step-size satisfies the Robbins-Monro condition. Hence, by proving the asymptotic stability of the O.D.E., we can induce the convergence of the original algorithm. In this paper, we will use an O.D.E. model of Q-learning, which is expressed as a special nonlinear system called a switching system.

## A.2  Assumption for Borkar and Meyn Theorem

**Assumption A.2.**
*1. The mapping $f : \mathbb{R}^n \to \mathbb{R}^n$ is globally Lipschitz continuous, and there exists a function $f_\infty : \mathbb{R}^n \to \mathbb{R}^n$ such that*

$$\lim_{c \to \infty} \frac{f(cx)}{c} = f_\infty(x), \quad \forall x \in \mathbb{R}^n. \tag{19}$$

*2. The origin in $\mathbb{R}^n$ is an asymptotically stable equilibrium for the O.D.E. $\dot{x}_t = f_\infty(x_t)$.*

*3. There exists a unique globally asymptotically stable equilibrium $\theta^e \in \mathbb{R}^n$ for the O.D.E. $\dot{x}_t = f(x_t)$, i.e., $x_t \to \theta^e$ as $t \to \infty$.*

*4. The sequence $\{m_k, \mathcal{G}_k\}_{k \geq 1}$ where $\mathcal{G}_k$ is sigma-algebra generated by $\{(\theta_i, m_i, k \geq i\}$, is a Martingale difference sequence. In addition , there exists a constant $C_0 < \infty$ such that for any initial $\theta_0 \in \mathbb{R}^n$ , we have $\mathbb{E}[||m_{k+1}||^2 | \mathcal{G}_k] \leq C_0(1 + ||\theta_k||^2), \forall k \geq 0$.*

*5. The step-sizes satisfies the Robbins-Monro condition [Robbins and Monro, 1951] :*

$$\sum_{k=0}^{\infty} \alpha_k = \infty, \quad \sum_{k=0}^{\infty} \alpha_k^2 < \infty.$$

## A.3  Auxiliary lemmas

**Lemma A.3** (Woodbury matrix identity [Hager, 1989]). *For $A, B \in \mathbb{R}^{n \times n}$, suppose $A$ and $I + A^{-1}B$ is invertible, then $A + B$ is invertible and we have*

$$(A + B)^{-1} = A^{-1} - A^{-1}B(I + A^{-1}B)^{-1}A^{-1}.$$

**Lemma A.4** (Gelfand's formula, Corollay 5.6.14 in Horn and Johnson [2013]). *For any matrix norm $|| \cdot ||$, for $A \in \mathbb{R}^{n \times n}$, we have*

$$\rho(A) = \lim_{k \to \infty} ||A^k||^{\frac{1}{k}},$$

*where $\rho(\cdot)$ denotes the spectral radius of a given matrix.*

**Definition A.5** (Theorem 3 in Molchanov and Pyatnitskiy [1989]). *A matrix $A \in \mathbb{R}^{n \times n}$ is said to have strictly negatively row dominating diagonal if $[A]_{ii} + \sum_{j \in \{1,2,\dots,n\} \setminus \{i\}} [A]_{ij} < 0$ for all $1 \leq i \leq n$.*

**Lemma A.6** (Theorem 3 in Molchanov and Pyatnitskiy [1989]). *Consider a switched system in (3) where $\mathcal{M} =: \{1, 2, \dots, M\}$ is the set for switching modes. If there exists a number $m \geq n$, a full-row rank matrix $L \in \mathbb{R}^{n \times m}$ and a set of matrices $\{\mathcal{L}_\sigma \in \mathbb{R}^{m \times m}\}_{\sigma \in \mathcal{M}}$ such that*

*1) Each $\mathcal{L}_\sigma$ for $\sigma \in \mathcal{M}$ has a strictly negatively row dominating diagonal:*

$$[\mathcal{L}]_{ii} + \sum_{j \in \{1,2,\dots,m\} \setminus \{i\}} |[\mathcal{L}]_{ij}| < 0.$$

*2) The following holds for all $\sigma \in \mathcal{M}$:*

$$A_\sigma^\top L = L \mathcal{L}_\sigma^\top.$$

*Then, the origin of (3) is asymptotically stable.*

**Lemma A.7** (Gerschgorin circle theorem [Horn and Johnson, 2013]). *Let $A \in \mathbb{R}^{n \times m}$ whose $i$-th row and $j$-th column element is $a_{ij}$. Let $R_i(A) = \sum_{j \in \{1,2,\dots,m\} \setminus \{i\}} a_{ij}$. Consider the Gerschgorin circles*

$$\{z \in \mathbb{C} : |z - a_{ii}| \leq R_i(A)\}, \quad i = 1, \dots, n.$$

*The eigenvalues of $A$ are in the union of Gerschgorin discs*

$$G(A) = \cup_{i=1}^n \{z \in \mathbb{C} : |z - a_{ii}| \leq R_i(A)\}.$$

Now, we state the lemma to guarantee positive definiteness of $A_{\pi_{X\theta}} + \eta I$. Instead we prove positive definiteness of $A_{\pi_{X\theta}} + \frac{\eta}{\lambda_{\max}(C)} C$. We follow the similar lines in Bharadwaj Diddigi et al. [2020].

**Lemma A.8.** *Let*

$$M^{\pi_{X\theta}} := D\left(\left(1 + \frac{\eta}{\lambda_{\max}(C)}\right) I - \gamma P^{\pi_{X\theta}}\right).$$

*Under the following condition:*

$$\eta > \lambda_{\max}(C) \max_{\substack{\pi \in \Theta \\ (s,a) \in \mathcal{S} \times \mathcal{A}}} \left(\frac{\gamma d^T P^{\pi_{X\theta}} (e_a \otimes e_s)}{2d(s,a)} - \frac{2 - \gamma}{2}\right),$$

*where $\Theta$ is the set of all deterministic policies, and $\otimes$ is the Kronecker product, $M^{\pi_{X\theta}}$ is positive definite.*

*Proof.* For simplicity of the notation, we will denote $d_i = d(s, a)$ and $e_i = e_a \otimes e_a$ for some $i \in \{1, 2, \dots, |\mathcal{S}||\mathcal{A}|\}$ where $i$ corresponds to some $s, a \in \mathcal{S} \times \mathcal{A}$.

We use Gerschgorin circle theorem for the proof. First, denote $m_{ij} = [M^{\pi_{X\theta}}]_{ij}$. Then, one gets

$$m_{ii} = d_i \left(\left(1 + \frac{\eta}{\lambda_{\max}(C)}\right) - \gamma e_i^T P^{\pi_{X\theta}} e_i\right),$$

$$m_{ij} = -d_i \gamma e_i^T P^{\pi_{X\theta}} e_j \quad \text{for} \quad i \neq j.$$

Except for the diagonal element, the row and column sums, respectively, become

$$\sum_{j \in S_i} |m_{ij}| = \gamma d_i (1 - e_i^T P^{\pi_{X\theta}} e_i),$$

$$\sum_{j \in S_i} |m_{ji}| = \gamma d^T P^{\pi_{X\theta}} e_i - \gamma d_i e_i^T P^{\pi_{X\theta}} e_i,$$

where $S_i = \{1, 2, \ldots, |\mathcal{S}||\mathcal{A}|\} \setminus \{i\}$. We need to show that $M^{\pi_{X\theta}} + M^{\pi_{X\theta}^T}$ is positive definite. To this end, we use Lemma A.7 to have the following inequality:

$$|\lambda - 2m_{ii}| \leq \sum_{j \in S_i} |m_{ij}| + \sum_{j \in S_i} |m_{ji}|.$$

Considering the lower bound of $\lambda$, we have

$$\lambda \geq 2m_{ii} - \sum_{j \in S_i} |m_{ij}| - \sum_{j \in S_i} |m_{ji}|$$

$$= 2d_i \left( \left( 1 + \frac{\eta}{\lambda_{\max}(C)} \right) - \gamma e_i^T P^{\pi_{X\theta}} e_i \right) - \gamma d_i (1 - e_i^T P^{\pi_{X\theta}} e_i) - (\gamma d^T P^{\pi_{X\theta}} e_i - \gamma d_i e_i^T P^{\pi_{X\theta}} e_i)$$

$$= \eta \frac{2d_i}{\lambda_{\max}(C)} + (2 - \gamma) d_i - \gamma d^T P^{\pi_{X\theta}} e_i.$$

Hence, for $\lambda > 0$, we should have

$$\eta > \lambda_{\max}(C) \left( \frac{\gamma d^T P^{\pi_{X\theta}} e_i}{2d_i} - \frac{2 - \gamma}{2} \right).$$

Taking $\eta > \lambda_{\max}(C) \max_{\substack{\pi \in \Theta \\ i \in \{1,2,\ldots,|\mathcal{S}||\mathcal{A}|\}}} \left( \frac{\gamma d^T P^{\pi_{X\theta}} e_i}{2d_i} - \frac{2-\gamma}{2} \right)$, we can make $M^{\pi_{X\theta}}$ always positive definite. This completes the proof. $\qquad\square$

We first introduce a lemma to bound the inverse of a matrix norm:

**Lemma A.9.** *[Page 351 in Horn and Johnson [2013]] If $M \in \mathbb{R}^{n \times n}$ satisfies $||M|| < 1$ for some matrix norm $|| \cdot ||$, then $I - M$ is non-singular, and*

$$\left\| (I - M)^{-1} \right\| \leq \frac{1}{1 - \|M\|}.$$

**Lemma A.10.** *Suppose that*

$$\left\| X^\top D X \right\|_\infty < \eta.$$

*Then, we have*

$$\left\| (X^\top D X + \eta I)^{-1} \right\|_\infty \leq \frac{1}{\eta - \|X^\top D X\|_\infty}.$$

*Proof.* We have

$$\left\| (X^\top D X + \eta I)^{-1} \right\|_\infty = \left\| \frac{1}{\eta} \left( \frac{1}{\eta} X^\top D X + I \right)^{-1} \right\|_\infty$$

$$= \frac{1}{\eta} \left\| \left( \frac{1}{\eta} X^\top D X + I \right)^{-1} \right\|_\infty$$

$$\leq \frac{1}{\eta} \frac{1}{1 - \left\| \frac{1}{\eta} X^\top D X \right\|_\infty}$$

$$= \frac{1}{\eta - \|X^\top D X\|_\infty}.$$

The first inequality follows from Lemma A.9. This completes the proof. $\qquad\square$

**Lemma A.11.** *For $\eta > \gamma \|X^\top D\|_\infty \|X\|_\infty + \|X^\top D X\|_\infty$, we have*

$$\gamma \left\| \Gamma_\eta \right\|_\infty < 1.$$

*Proof.* From the definition of $\Gamma_\eta$ in (10), we have

$$
\begin{aligned}
\gamma \left\|\Gamma_\eta\right\|_\infty =& \gamma \left\|X^\top D(X^\top DX + \eta I)^{-1} X\right\|_\infty \\
\leq& \gamma \left\|X^\top D\right\|_\infty \|X\|_\infty \frac{1}{\eta - \|X^\top DX\|_\infty} \\
<& 1.
\end{aligned}
$$

The first inequality follows from Lemma A.10. The last inequality follows from the condition $\eta > \gamma\|X^\top D\|_\infty\|X\|_\infty + \|X^\top DX\|_\infty$. This completes the proof. $\square$

**Lemma A.12.** *The equation (7) can be written as*

$$
X\theta_\eta^* = \Gamma_\eta \mathcal{T} X\theta_\eta^*.
$$

*Proof.* Let us expand the terms in (7):

$$
\begin{aligned}
X^\top DR =& (\eta I + X^\top DX)\theta_\eta^* - \gamma X^\top DP\Pi_{X\theta_\eta^*} X\theta_\eta^* \\
\iff& X^\top D(R + \gamma P\Pi_{X\theta_\eta^*} X\theta_\eta^*) = (\eta I + X^\top DX)\theta_\eta^* \\
\iff& X(\eta I + X^\top DX)^{-1} X^\top D(R + \gamma P\Pi_{X\theta_\eta^*} X\theta_\eta^*) = X\theta_\eta^*.
\end{aligned}
$$

The last line follows from that $X$ is full-column rank matrix. This completes the proof. $\square$

**Lemma A.13.** *For any $\theta \in \mathbb{R}^h$, if $\eta > \gamma\|X^\top D\|_\infty\|X\|_\infty + \|X^\top DX\|_\infty$, then $-A_{\pi_{X\theta}} - \eta I$ has strictly negatively row dominating diagonal.*

*Proof.* For $1 \leq i \leq h$, we have

$$
\begin{aligned}
[-A_{\pi_{X\theta}} - \eta I]_{ii} + \sum_{j\in\{1,2,\dots,h\}\setminus\{i\}} |[A_{\pi_{X\theta}} + \eta I]_{ij}| \leq& -\eta + \sum_{j=1}^h |[A_{\pi_{X\theta}}]_{ij}| \\
\leq& -\eta + \|A_{\pi_{X\theta}}\|_\infty \\
\leq& -\eta + \left\|X^\top DX\right\|_\infty + \gamma \left\|X^\top D\right\|_\infty \|X\|_\infty \\
<& 0.
\end{aligned}
$$

The second last inequality follows the fact that $\|P\Pi_{X_\theta}\|_\infty \leq 1$. $\square$

**Lemma A.14** (Continuity of $\theta_\eta^*$ with respect to $\eta$). *Let $\eta_0$ be a non-negative real valued constant. Suppose $\gamma\|\Gamma_{\eta_0}\|_\infty < 1$. Then, $\theta_\eta^*$ is continuous at $\eta_0$.*

*Proof.* Note that $\Gamma_\eta$ is continuous function of $\eta$, and we have,

$$
\Gamma_{\eta_0+\eta} = \Gamma_{\eta_0} + O(\eta),
$$

where $O(\cdot)$ stands for the big O notation. Therefore,

$$
\begin{aligned}
\|X\theta_{\eta_0+\eta}^* - X\theta_{\eta_0}^*\|_\infty =& \|\Gamma_{\eta_0+\eta}\mathcal{T}X\theta_{\eta_0+\eta}^* - \Gamma_{\eta_0}\mathcal{T}X\theta_{\eta_0}^*\|_\infty \\
\leq& \left\|\Gamma_{\eta_0}\mathcal{T}X(\theta_{\eta_0+\eta}^* - \theta_{\eta_0}^*)\right\|_\infty + O(\eta) \\
\leq& \gamma \left\|\Gamma_{\eta_0}\right\|_\infty \|X(\theta_{\eta_0+\eta}^* - \theta_{\eta_0}^*)\|_\infty + O(\eta).
\end{aligned}
$$

The first equality follows from the definition of $\theta_{\eta_0+\eta}^*$ and $\theta_{\eta_0}^*$. The second inequality follows from triangle inequality. The last inequality follows from the contraction property of the Bellman operator. Therefore, we have

$$
\|\theta_{\eta_0+\eta}^* - \theta_{\eta_0}^*\|_\infty \leq C\|X\theta_{\eta_0+\eta}^* - X\theta_{\eta_0}^*\|_\infty \leq O(\eta),
$$

where the first inequality holds because $X$ is full-column rank matrix, and $C$ is a universal constant. This completes the proof. $\square$

## A.4 Proof of Lemma 2.5

*Proof.* The first item follows from Lemma A.6. The second item follows from the fact that if all the subsystem matrices are negative-definite, then it will have $V(x) = ||x||_2^2$ as a common Lyapunov function. This completes the proof. □

## A.5 Proof of Lemma 3.1

*Proof.* Let us prove the first item. When, $\eta \to 0$ , we have $(X^\top DX + \eta I)^{-1} \to (X^\top DX)^{-1}$. Therefore, we have $\Gamma_\eta \to \Gamma$ as $\eta \to 0$.

Moreover, note that from Lemma A.10, for sufficiently large $\eta$, we have $(X^\top DX + \eta I)^{-1}$ is invertible. Therefore, we get

$$\|\Gamma_\eta\|_\infty = \|X^\top D(X^\top DX + \eta I)^{-1} X\|_\infty$$
$$\leq \|X^\top D\|_\infty \|X\|_\infty \frac{1}{\eta - \|X^\top DX\|_\infty},$$

where the first inequality follows from Lemma A.10. As $\eta \to \infty$, we get $\|\Gamma_\eta\|_\infty \to 0$. This completes the proof of the first item.

Now, we will prove the second item. First of all, using Woodbury matrix identity [Hager, 1989] in Lemma A.3, we have

$$(X^\top DX + \eta I)^{-1} = (X^\top DX)^{-1} - (X^\top DX + \eta^{-1}(X^\top DX)(X^\top DX))^{-1}$$
$$\preceq (X^\top DX)^{-1},$$

where the inequality comes from the fact that $(X^\top DX + \eta^{-1}(X^\top DX)(X^\top DX))^{-1}$ is positive semidefinite. Then, we have

$$\|\Gamma_\eta\|_\infty = \|X^\top D(X^\top DX + \eta I)^{-1} X\|_\infty$$
$$= \sqrt{|S \times A|} \|X^\top D(X^\top DX + \eta I)^{-1} X\|_2$$
$$\leq \sqrt{|S \times A|} \|X^\top D\|_2 \|X\|_2 \|(X^\top DX + \eta I)^{-1}\|_2.$$

Next, since the spectral norm is monotone, for any two symmetric positive semidefinite matrices $A$ and $B$, $A \succeq B$ implies $\|A\|_2 \geq \|B\|_2$, which comes from the properties of the spectral norm for symmetric positive semidefinite matrices. Therefore, one gets

$$\|\Gamma_\eta\|_\infty \leq \|X^\top D\|_2 \|X\|_2 \|(X^\top DX + \eta I)^{-1}\|_2 \sqrt{|S \times A|}$$
$$\leq \|X^\top D\|_2 \|X\|_2 \|(X^\top DX)^{-1}\|_2 \sqrt{|S \times A|},$$

which is the desired conclusion.

□

## A.6 Proof of Lemma 3.2

*Proof.* To show the existence and uniqueness of the solution of (9), we use Banach fixed-point theorem. Note that it is enough show the existence and uniqueness of the solution of the following equation:

$$y = \Gamma_\eta(R + \gamma P \Pi_y y), \quad y \in \mathbb{R}^h. \tag{20}$$

This is because a solution $y^* \in \mathbb{R}^h$ satisfying the above equation is in the image of $X$. We can find a unique $\theta$ such that $X\theta = y^*$ because $X$ is a full-column rank matrix. To this end, we will apply the Banach fixed point theorem:

$$\|y_1 - y_2\|_\infty = \|X(X^T DX + \eta I)^{-1}(\gamma X^T DP\Pi_{y_1} y_1 - \gamma X^T DP\Pi_{y_2} y_2)\|_\infty$$
$$\leq \gamma \|X(X^T DX + \eta I)^{-1} X^\top D\|_\infty \|\Pi_{y_1} y_1 - \Pi_{y_2} y_2\|_\infty$$
$$\leq \gamma \|X(X^T DX + \eta I)^{-1} X^\top D\|_\infty \|y_1 - y_2\|_\infty$$
$$< \|y_1 - y_2\|_\infty.$$

The second inequality follows from the non-expansiveness property of the max-operator. Now, we can use Banach fixed-point theorem to conclude existence and uniqueness of (20). This completes the proof. □

## A.7 Proof of Lemma 3.4

For the proof, suppose that $\gamma\|\Gamma\|_\infty < 1$. If the condition $0 \leq \eta < \frac{(1-\gamma\|\Gamma\|_\infty)\|(X^TDX)^{-1}\|_\infty^{-1}}{\gamma\|(X^TDX)^{-1}\|_\infty\|X\|_\infty\|X^TD\|_\infty+(1-\gamma\|\Gamma\|_\infty)}$ holds, it ensures $\left\|\eta(X^TDX)^{-1}\right\|_\infty < 1$ since $\frac{(1-\gamma\|\Gamma\|_\infty)}{\gamma\|(X^TDX)^{-1}\|_\infty\|X\|_\infty\|X^TD\|_\infty+(1-\gamma\|\Gamma\|_\infty)} < 1$. Then, using Gelfand's formula in Lemma A.4, we can easily prove that the spectral radius of $\eta(X^TDX)^{-1}$ is less than one. Next, note that for any two square matrices $A$ and $B$, $(A-B)^{-1} = \sum_{i=0}^{\infty}(A^{-1}B)^iA^{-1}$ if the spectral radius of $A^{-1}B$ is less than one. Using this fact, one has

$$
\begin{aligned}
\gamma\left\|\Gamma_\eta\right\|_\infty &= \left\|\gamma X(X^TDX + \eta I)^{-1}X^TD\right\|_\infty \\
&= \gamma\left\|X\sum_{i=0}^{\infty}(-\eta(X^TDX)^{-1})^i(X^TDX)^{-1}X^TD\right\|_\infty \\
&\leq \gamma\left\|X(X^TDX)^{-1}X^TD\right\|_\infty + \gamma\left\|(X^TDX)^{-1}\sum_{i=1}^{\infty}\eta^iX(-X^TDX)^{-i}X^TD\right\|_\infty \\
&\leq \gamma\left\|X(X^TDX)^{-1}X^TD\right\|_\infty + \gamma\eta\left\|(X^TDX)^{-1}\right\|_\infty^2\|X\|_\infty\|X^TD\|_\infty\sum_{i=0}^{\infty}\left\|\eta(X^TDX)^{-1}\right\|_\infty^i \\
&\leq \gamma\|\Gamma\|_\infty + \frac{\gamma\eta\left\|(X^TDX)^{-1}\right\|_\infty^2\|X\|_\infty\|X^TD\|_\infty}{1-\eta\|(X^TDX)^{-1}\|_\infty},
\end{aligned}
$$

where the second line uses the matrix inverse property. Therefore, $\gamma\left\|\Gamma_\eta\right\|_\infty < 1$ holds if

$$
\gamma\|\Gamma\|_\infty + \frac{\gamma\eta\left\|(X^TDX)^{-1}\right\|_\infty^2\|X\|_\infty\|X^TD\|_\infty}{1-\eta\left\|(X^TDX)^{-1}\right\|_\infty} < 1.
$$

Rearranging terms, one gets the desired conclusion.

## A.8 Proof of Lemma 3.5

From the definition of $\Gamma_\eta$ in (10), we have

$$
\begin{aligned}
\gamma\left\|\Gamma_\eta\right\|_\infty &\leq \gamma\|X\|_\infty\left\|(X^\top DX + \eta I)^{-1}\right\|_\infty\left\|X^\top\right\|_\infty\|D\|_\infty \\
&\leq \gamma\frac{1}{|\mathcal{S}||\mathcal{A}|}\frac{1}{a+\eta} \\
&< 1.
\end{aligned}
$$

The second inequality follows from the assumption that $X^\top DX = aI$ and $\|X\|_2 \leq 1$. The last inequality follows from the condition $a|\mathcal{S}||\mathcal{A}| \geq 1$.

## A.9 Proof of Lemma 3.6

*Proof.* Since we are going to consider the case $\eta \to \infty$, assume that $\eta > \left\|X^\top DX\right\|_\infty + \gamma\left\|X^\top D\right\|_\infty\|X\|_\infty$. From (7), we have

$$
\begin{aligned}
\left\|\theta_\eta^*\right\|_\infty &= \left\|(X^\top DX + \eta I)^{-1}(X^\top DR + \gamma X^\top DP\Pi_{X\theta_\eta^*}X\theta_\eta^*)\right\|_\infty \\
&\leq \frac{1}{\eta - \|X^\top DX\|_\infty}\left\|X^\top DR + \gamma X^\top DP\Pi_{X\theta_\eta^*}X\theta_\eta^*\right\|_\infty \\
&\leq \frac{1}{\eta - \|X^\top DX\|_\infty}\left\|X^\top DR\right\|_\infty + \frac{1}{\eta - \|X^\top DX\|_\infty}\left\|X^\top D\right\|_\infty\|X\|_\infty\left\|\theta_\eta^*\right\|_\infty.
\end{aligned}
$$

The first inequality follows from Lemma A.10. Therefore, considering that $\eta > \left\|X^\top D X\right\|_\infty + \gamma \left\|X^\top D\right\|_\infty \|X\|_\infty$, we have

$$\frac{\eta - \left\|X^\top D X\right\|_\infty - \gamma \left\|X^\top D\right\|_\infty \|X\|_\infty}{\eta - \|X^\top D X\|_\infty} \left\|\theta_\eta^*\right\|_\infty < \frac{1}{\eta - \|X^\top D X\|_\infty} \left\|X^\top D R\right\|_\infty,$$

which leads to

$$\left\|\theta_\eta^*\right\|_\infty \leq \frac{1}{\eta - \|X^\top D X\|_\infty - \gamma \|X^\top D\|_\infty \|X\|_\infty} \left\|X^\top D R\right\|_\infty.$$

As $\eta \to \infty$, the right-hand side of the above equation goes to zero, i.e., $\theta_\eta^* \to 0$. $\qquad\square$

## A.10 Proof of Lemma 3.7

*Proof.* The bias term of the solution can be obtained using simple algebraic inequalities.

$$\begin{aligned}
\|X\theta_\eta^* - Q^*\|_\infty &\leq \left\|\Gamma_\eta \mathcal{T}(X\theta_\eta^*) - \Gamma Q^*\right\|_\infty + \|\Gamma_\eta Q^* - Q^*\|_\infty \\
&\leq \|\Gamma_\eta\|_\infty \left\|\mathcal{T}(X\theta_\eta^*) - Q^*\right\|_\infty + \|\Gamma_\eta Q^* - Q^*\|_\infty \\
&= \|\Gamma_\eta\|_\infty \left\|\mathcal{T}(X\theta_\eta^*) - \mathcal{T}(Q^*)\right\|_\infty + \|\Gamma_\eta Q^* - Q^*\|_\infty \\
&\leq \gamma \|\Gamma_\eta\|_\infty \left\|X\theta_\eta^* - Q^*\right\|_\infty + \|\Gamma_\eta Q^* - Q^*\|_\infty.
\end{aligned}$$

The first inequality follows from triangle inequality. The third equality follows from the fact that $Q^*$ is the solution of optimal Bellman equation. The last inequality follows from the contraction property of the Bellman operator. Noting that $\gamma \|\Gamma_\eta\|_\infty < 1$, we have

$$\left\|X\theta_\eta^* - Q^*\right\|_\infty \leq \frac{1}{1 - \gamma \|\Gamma_\eta\|_\infty} \|\Gamma_\eta Q^* - Q^*\|_\infty.$$

This finishes the proof.

$\qquad\square$

## A.11 Proofs to check Assumption A.2 for Theorem 5.2.

In this section, we provide omitted proofs to check Assumption A.2 in Appendix Section A.2 to apply the Borkar and Meyn Theorem in Lemma A.1 in the Appendix.

First of all, Lipschitzness of $f(\theta)$ ensures the unique solution of the O.D.E..

**Lemma A.15** (Lipschitzness). *Let*

$$f(\theta) = -(X^T D X + \eta I)\theta + \gamma X^T D P \Pi_{X\theta} X \theta + X^T D R. \tag{21}$$

*Then, $f(\theta)$ is globally Lipschitzness continuous.*

*Proof.* Lipschitzness of $f(\theta)$ can be proven as follows:

$$\begin{aligned}
||f(\theta) - f(\theta')||_\infty &\leq ||(X^T D X + \eta I)(\theta - \theta')||_\infty + \gamma ||X^T D P (\Pi_{X\theta} X\theta - \Pi_{X\theta'} X\theta')||_\infty \\
&\leq ||X^T D X + \eta I||_\infty ||\theta - \theta'||_\infty + \gamma ||X^T D P||_\infty ||\Pi_{X\theta} X\theta - \Pi_{X\theta'} X\theta'||_\infty \\
&\leq (||X^T D X + \eta I||_\infty + \gamma ||X^T D P||_\infty ||X||_\infty))||\theta - \theta'||_\infty
\end{aligned}$$

The last inequality follows from non-expansiveness property of max-operator. Therefore $f(\theta)$ is Lipschitz continuous with respect to the $||\cdot||_\infty$, $\qquad\square$

Next, the existence of limiting O.D.E. of (16) can be proved using the fact that policy is invariant under constant multiplication when linear function approximation is used.

**Lemma A.16** (Existence of limiting O.D.E. and stability). *Let*

$$f(\theta) = (-X^T D X - \eta I)\theta + \gamma X^T D P \Pi_{X\theta} X \theta + X^T D R. \tag{22}$$

*If $\eta$ satisfies (15), there exists limiting O.D.E. of (22) and its origin is asymptotically stable.*

*Proof.* The existence of limiting O.D.E. can be obtained using the homogeneity of policy, $\Pi_{X(c\theta)} = \Pi_{X\theta}$.

$$f(c\theta) = -(X^T D X + \eta I)(c\theta) + \gamma X^T D P \Pi_{X(c\theta)} X(c\theta) + X^T D R,$$

$$\lim_{c\to\infty} \frac{f(cx)}{c} = (-X^T D X - \eta I + \gamma X^T D P \Pi_{X\theta} X)\theta$$

This can be seen as switching system and, from Lemma A.8 and Lemma A.13, we can apply Lemma 2.5. Therefore, the origin is asymptotically stable. $\square$

Lastly, we check conditions for martingale difference sequences.

**Lemma A.17** (Martingale difference sequence, $m_k$, and square integrability)**.** *We have*

$$\mathbb{E}[m_{k+1}|\mathcal{F}_k] = 0,$$
$$\mathbb{E}[||m_{k+1}||_2^2|\mathcal{F}_k] < C_0(1 + ||\theta_k||_2^2),$$

*where* $C_0 := \max(12 X_{\max}^2 R_{\max}^2, 12\gamma X_{\max}^4 + 4\eta^2)$.

*Proof.* To show $\{m_k, k \in \mathbb{N}\}$ is a martingale difference sequence with respect to the sigma-algebra generated by $\mathcal{G}_k$, we first prove expectation of $m_{k+1}$ is zero conditioned on $\mathcal{G}_k$:

$$\mathbb{E}[m_{k+1}|\mathcal{G}_k] = 0.$$

This follows from definition of $b, C$ and $A_{\pi_{X\theta}}$.

The boundedness $\mathbb{E}[||m_k||_2] < \infty$ for $k \in \mathbb{N}$ also follows from simple algebraic inequalities. Therefore $\{m_k, k \in \mathbb{N}\}$ is martingale difference sequence.

Now, we show that the following hods:

$$\mathbb{E}[||m_{k+1}||_2^2|\mathcal{G}_k] \leq C_0(||\theta_k||_2^2 + 1).$$

Using simple algebraic inequalities, we have

$$
\begin{aligned}
\mathbb{E}[||m_{k+1}||_2^2|\mathcal{G}_k] &= \mathbb{E}[||\delta_k x(s_k, a_k) + \eta\theta_k - \mathbb{E}_\mu[\delta_k x(s_k, a_k) + \eta\theta_k]||_2^2|\mathcal{G}_t] \\
&\leq \mathbb{E}[||\delta_k x(s_k, a_k) + \eta\theta_k||_2^2 + ||\mathbb{E}_\mu[\delta_k x(s_k, a_k) + \eta\theta_k]||_2^2|\mathcal{G}_t] \\
&\leq 2\mathbb{E}[||\delta_k x(s_k, a_k) + \eta\theta_k||_2^2|\mathcal{G}_t] \\
&\leq 4\mathbb{E}[||\delta_k x(s_k, a_k)||_2^2|\mathcal{G}_t] + 4\eta^2 \mathbb{E}[||\theta_k||_2^2|\mathcal{G}_t] \\
&\leq 12 X_{\max}^2 \mathbb{E}[||r_k||_2^2 + ||\gamma \max x(s_k, a_k)\theta_k||_2^2 + ||x(s_k, a_k)\theta_k||_2^2|\mathcal{G}_t] + 4\eta^2||\theta_k||_2^2 \\
&\leq 12 X_{\max}^2 R_{\max}^2 + 12\gamma X_{\max}^4 ||\theta_k||_2^2 + ||\theta_k||_2^2 + 4\eta^2||\theta_k||_2^2 \\
&\leq C_0(1 + ||\theta_k||_2^2),
\end{aligned}
$$

where $C_0 := \max(12 X_{\max}^2 R_{\max}^2, 12\gamma X_{\max}^4 + 4\eta^2)$. The fourth inequality follows from the fact that $||a + b + c||_2^2 \leq 3||a||_2^2 + 3||b||_2^2 + 3||c||_2^2$. Together with the above inequality, the square integrability of $m_k$ for $k \in \mathbb{N}$ follows from the recursive update of $\theta_k$ in (14). This completes the proof.

$\square$

## A.12 Proof of Theorem 5.1

Before moving onto the proof of Theorem 5.1, in order to prove the stability using the upper and lower systems, we need to introduce some notions such as the quasi-monotone function and vector comparison principle. We first introduce the notion of quasi-monotone increasing function, which is a necessary prerequisite for the comparison principle for multidimensional vector system.

**Definition A.18** (Quasi-monotone function)**.** *Consider a vector-valued function* $f : \mathbb{R}^n \to \mathbb{R}^n$ *with* $f := [f_1 \quad f_2 \quad \cdots \quad f_n]^T$ *where* $f_i : \mathbb{R}^n \to \mathbb{R}$ *for* $i \in \{1, 2, \dots, n\}$. $f$ *is said to be quasi-monotone increasing if* $f_i(x) \leq f_i(y)$ *holds for all* $i \in \{1, 2, \dots, n\}$ *and* $x, y \in \mathbb{R}^n$ *such that* $x_i = y_i$ *and* $x_j \leq y_j$ *for all* $j \neq i$.

Based on the notion of quasi-monotone function, we introduce the vector comparison principle.

**Lemma A.19** (Vector Comparison Principle, Hirsch and Smith [2006]). *Suppose that $\bar{f}, \underline{f}$ are globally Lipschitz continuous. Let $x_t$ be a solution of the system*

$$\frac{d}{dt}x_t = \bar{f}(x_t), \quad x_0 \in \mathbb{R}^n, \quad \forall t \geq 0.$$

*Assume that $\bar{f}$ is quasi-monotone increasing, and let $v_t$ be a solution of the system*

$$\frac{d}{dt}v_t = \underline{f}(v_t), \quad v_0 < x_0, \quad \forall t \geq 0,$$

*where $\underline{f}(v) \leq \bar{f}(v)$ holds for any $v \in \mathbb{R}^n$. Then, $v_t \leq x_t$ for all $t \geq 0$.*

The vector comparison lemma can be used to bound the state trajectory of the original system by those of the upper and lower systems. Then, proving global asymptotic stability of the upper and lower systems leads to global asymptotic stability of original system. We now give the proof of Theorem 5.1.

*Proof.* First we construct the upper comparison part. Noting that

$$\gamma X^T DP\Pi_{X\theta_\eta^*} X\theta_\eta^* \geq \gamma X^T DP\Pi_{X\theta} X\theta_\eta^* \tag{23}$$

and

$$\gamma X^T DP\Pi_{X(\theta-\theta_\eta^*)} X(\theta - \theta_\eta^*) \geq \gamma X^T DP\Pi_{X\theta} X(\theta - \theta_\eta^*), \tag{24}$$

we define $\bar{f}(y)$ and $\underline{f}(y)$ as follows:

$$\bar{f}(y) = (-X^T DX - \eta I + \gamma X^T DP\Pi_{Xy} X)y,$$
$$\underline{f}(y) = (-X^T DX - \eta I + \gamma X^T DP\Pi_{X(y+\theta_\eta^*)} X)y + \gamma X^T DP(\Pi_{X(y+\theta_\eta^*)} - \Pi_{X\theta_\eta^*})X\theta_\eta^*.$$

Using (23) and (24), we have $\underline{f}(y) \leq \bar{f}(y)$.

$\underline{f}$ is the corresponding O.D.E. of original system in (17) and $\bar{f}$ becomes O.D.E. of the upper system. $\bar{f}$ becomes switched linear system.

Now, consider the O.D.E. systems

$$\frac{d}{dt}\theta_t^u = \bar{f}(\theta_t^u), \qquad \theta_0^u > \theta_0,$$
$$\frac{d}{dt}\theta_t = \underline{f}(\theta_t).$$

Next, we prove quasi-monotone increasing property of $\bar{f}$. For any $y \in \mathbb{R}^h$, consider a non-negative vector $p \in \mathbb{R}^h$ such that its $i$-th element is zero. Then, for any $1 \leq i \leq h$, we have

$$\begin{aligned}
e_i^T \bar{f}(y+p) &= e_i^T(-X^T DX - \eta I + \gamma X^T DP\Pi_{X(y+p)} X)(y+p) \\
&= -e_i^T(X^T DX + \eta I)y - \eta e_i^T p + \gamma e_i^T X^T DP\Pi_{X(y+p)} X(y+p) \\
&\geq -e_i^T(X^T DX + \eta I)y + \gamma e_i^T X^T DP\Pi_{Xy} Xy \\
&= e_i^T \bar{f}(y),
\end{aligned}$$

where the inequality comes from $e_i^T X^T DXp = 0$ due to Assumption 2.2 and $e_i^T p = 0$ since $i$-th element of $p$ is zero.

Therefore by Lemma A.19, we can conclude that $\theta_t \leq \theta_t^u$. The condition $(S1)$ in (15) ensures the switching matrices to have strictly negatively row dominating diagonal. Therefore, from Lemma A.13, and from Lemma 2.5, the global asymptotically stability of the origin follows. Likewise, the condition $(S2)$ in (15) ensures that the matrices are all negative-definite, implying that the switching system shares $V(\theta) = ||\theta||_2^2$ as common Lyapunov function. Therefore, we can conclude that the upper comparison system is globally asymptotically stable.

For the lower comparison part, noting that

$$\gamma X^T DP\Pi_{X\theta} X\theta \geq \gamma X^T DP\Pi_{X\theta_\eta^*} X\theta,$$

we can define $\underline{f}(y)$ and $\bar{f}(y)$ such that $\underline{f}(y) \leq \bar{f}(y)$ as follows:
$$\bar{f}(y) = -X^T DXy - \eta y + \gamma X^T DP\Pi_{Xy} Xy + X^T DR,$$
$$\underline{f}(y) = -X^T DXy - \eta y + \gamma X^T DP\Pi_{X\theta_\eta^*} Xy + X^T DR.$$

The corresponding O.D.E. system becomes
$$\frac{d}{dt}\theta_t = \bar{f}(\theta_t),$$
$$\frac{d}{dt}\theta_t^l = \underline{f}(\theta_t^l), \quad \theta_0^l < \theta_0. \tag{25}$$

Proving the quasi-monotonicity of $\bar{f}$ is similar to previous step. Consider a non-negative vector $p \in \mathbb{R}^h$ such that its $i$-th element is zero. Then, we have
$$\begin{aligned}
e_i^T \bar{f}(y+p) &= e_i^T(-(X^T DX + \eta I)(y+p) + \gamma X^T DP\Pi_{X(y+p)} X(y+p) + X^T DR)\\
&= e_i^T(-(X^T DX + \eta I)y + \gamma X^T DP\Pi_{X(y+p)} X(y+p) + X^T DR)\\
&\geq e_i^T(-(X^T DX + \eta I)y + \gamma X^T DP\Pi_{Xy} Xy + X^T DR)\\
&= e_i^T \bar{f}(y).
\end{aligned}$$

The second equality holds since $X^T DX$ is diagonal matrix and $p_i = 0$. Therefore by Lemma A.19, we can conclude that $\theta_t^l \leq \theta_t$. The lower comparison part is linear system without affine term. Hence, following the similar lines as in proving the stability of upper comparison system, we can conclude that (25) is globally asymptotically stable.

To prove uniqueness of the equilibrium point, assume there exists two different equilibrium points $\theta_1^e$ and $\theta_2^e$. The global asymptotic stability implies that regardless of initial state, $\theta_t \to \theta_1^e$ and $\theta_t \to \theta_2^e$. However this becomes contradiction if $\theta_1^e \neq \theta_2^e$. Therefore, the equilibrium point is unique. □

### A.13 Proof of Theorem 5.2

*Proof.* To apply Lemma A.1, let us check Assumption A.2.

1. First and second statement of Assumption A.2 follows from Lemma A.16.

2. Third statement of Assumption A.2 follows from Theorem 5.1.

3. Fourth statement of Assumption A.2 follows from Lemma A.17.

Since we assumed Robbins Monro step-size, we can now apply Lemma A.1 to complete the proof. □

### A.14 Example for non-existence of solution of PBE

Let us define a MDP whose state transition diagram is given as in Figure 3. The cardinality of state space and action space are $|\mathcal{S}| = 3$, $|\mathcal{A}| = 2$ respectively. The corresponding state transition matrix, and other parameters are given as follows:

$$X = \begin{bmatrix} 1 & 0 \\ 2 & 0 \\ 0 & 1 \\ 0 & 1 \\ 2 & 0 \\ 0 & 1 \end{bmatrix}, \ R_1 = \begin{bmatrix} -2 \\ 0 \\ 0 \end{bmatrix}, \ R_2 = \begin{bmatrix} 1 \\ 0 \\ 0 \end{bmatrix},$$

$$P_1 = \begin{bmatrix} 0 & 1 & 0 \\ \frac{1}{4} & \frac{1}{4} & \frac{1}{2} \\ \frac{1}{4} & \frac{1}{2} & \frac{1}{4} \end{bmatrix}, \ P_2 = \begin{bmatrix} 0 & 0 & 1 \\ \frac{1}{4} & \frac{1}{4} & \frac{1}{2} \\ \frac{1}{4} & \frac{1}{2} & \frac{1}{4} \end{bmatrix},$$

$$\gamma = 0.99, \quad d(s,a) = \frac{1}{6}, \ \forall s \in \mathcal{S}, \forall a \in \mathcal{A},$$

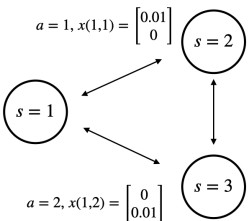

Figure 3: State transition diagram

where the order of elements of each column follows the orders of the corresponding definitions. Note that for this Markov decision process, taking action $a = 1$ and action $a = 2$ at state $s = 2$ have the same transition probabilities and reward. It is similar for the state $s = 3$. In this MDP, there are only two deterministic policies available, denoted by $\pi_1$ and $\pi_2$, that selects action $a = 1$ and action $a = 2$ at state $s = 1$, respectively, i.e., $\pi_1(1) = 1$ and $\pi_2(1) = 2$. The actions at state $s = 2$ and $s = 3$ do not affect the overall results.

The motivation of this MDP is as follows. Substitute $\pi_{X\theta^*}$ in (5) with $\pi_1$ and $\pi_2$. Then each of its solution becomes

$$\theta^{e1} := \begin{bmatrix} \theta_1^{e1} \\ \theta_2^{e1} \end{bmatrix} \approx \begin{bmatrix} -0.85 \\ -0.72 \end{bmatrix}, \quad \theta^{e2} := \begin{bmatrix} \theta_1^{e2} \\ \theta_2^{e2} \end{bmatrix} \approx \begin{bmatrix} -1.26 \\ -1.46 \end{bmatrix}.$$

If $\pi_1$ is the corresponding policy to the solution of (5), it means that action $a = 1$ is greedily selected at state $s = 1$. Therefore, $Q^{\pi_1}(1, 1) > Q^{\pi_1}(1, 2)$ should be satisfied. However, since $Q^{\pi_1}(1, 1) = x(1, 1)^T \theta^{e1} \approx -0.85$ and $Q^{\pi_1}(1, 2) = x(1, 2)^T \theta^{e1} \approx -0.72$, this is contradiction. The same logic applies to the case for $\pi_2$. Therefore, neither of them becomes a solution of (5). On the other hand, considering (7) with $\eta = 4$ which satisfies (15), the solution for each policy becomes $\theta_1^{e1} \approx -0.069$, $\theta_2^{e1} \approx 0.032$ and $\theta_1^{e2} \approx -0.069$, $\theta_2^{e2} \approx 0.035$, respectively. For $\pi_1$ and $\pi_2$, we have $Q^{\pi_1}(1, 1) < Q^{\pi_2}(1, 2)$ and $Q^{\pi_1}(1, 1) < Q^{\pi_1}(1, 2)$ respectively. Hence, $\theta^{e2}$ satisfies (7) and becomes the unique solution.

### A.15 Discussion on (15)

In this section, we provide further discussion on (15). The two conditions $(S1)$ and $(S2)$ are to make a matrix to have strictly negatively row dominating diagonal or negative definite, respectively. We first provide a case where a matrix with strictly negatively row dominating diagonal is not necessarily a negative definite matrix, and vice versa. We will consider MDPs with $\gamma = 0.99$.

*A matrix with strictly negatively row dominating diagonal but not negative-definite:* Consider the following MDP with only single action for each state:

$$X = \begin{bmatrix} 13 & -4 \\ 1 & 8 \end{bmatrix}, \quad P = \begin{bmatrix} 0 & 1 \\ 0 & 1 \end{bmatrix}, \quad D = \begin{bmatrix} \frac{1}{2} & 0 \\ 0 & \frac{1}{2} \end{bmatrix}, \quad \Pi = \begin{bmatrix} 1 & 0 \\ 0 & 1 \end{bmatrix}.$$

where the matrix $\Pi$ represents the policy. Then, we have

$$M := -X^\top DX + \gamma X^\top DP\Pi X \approx \begin{bmatrix} -78 & -77 \\ 24 & -24.2 \end{bmatrix}.$$

This is a matrix with strictly negatively row dominating diagonal but $M + M^\top$ is not negative definite matrix.

*A negative-definite matrix but not with a strictly negatively row dominating diagonal:* Consider the following MDP with single action for each state:

$$X = \begin{bmatrix} -1 & -4 \\ 0 & 5 \end{bmatrix}, \quad P = \begin{bmatrix} \frac{1}{2} & \frac{1}{2} \\ \frac{1}{2} & \frac{1}{2} \end{bmatrix}, \quad D = \begin{bmatrix} \frac{1}{2} & 0 \\ 0 & \frac{1}{2} \end{bmatrix}, \quad \Pi = \begin{bmatrix} 1 & 0 \\ 0 & 1 \end{bmatrix}.$$

Then, we have

$$M := -X^\top DX + \gamma X^\top DP\Pi X \approx \begin{bmatrix} -0.25 & -2.25 \\ -2.25 & -20 \end{bmatrix}.$$

$M + M^\top$ is negative definite matrix but $M$ does not have strictly negatively row dominating diagonal.

Now, we provide an example where the condition $(S1)$ and $(S2)$ in (15) does not imply each other, i.e., there are cases such that $(S1) \geq (S2)$ or $(S2) \geq (S1)$:

As for the condition in (15), consider the following MDP with single action for each state:

$$X = \begin{bmatrix} 1 \\ 2 \end{bmatrix}, \quad D = \begin{bmatrix} \frac{1}{100} & 0 \\ 0 & \frac{99}{100} \end{bmatrix}, \quad P_1 = \begin{bmatrix} \frac{1}{2} & \frac{1}{2} \\ \frac{1}{2} & \frac{1}{2} \end{bmatrix}, \quad P_2 = \begin{bmatrix} 0 & 1 \\ 0 & 1 \end{bmatrix}, \quad \Pi = \begin{bmatrix} 1 & 0 \\ 0 & 1 \end{bmatrix}.$$

where $P_1$ and $P_2$ are two different transition matrices and the matrix $\Pi$ represents the policy. First, considering $P_1$, we have

$$(S1) \approx 7.9, \quad (S2) \approx 196.$$

Meanwhile, considering $P_2$ as the transition matrix, we have

$$(S1) \approx 7.9, \quad (S2) \approx 2.$$

Therefore, $(S1)$ and $(S2)$ does not necessarily imply each other.

### A.16 Pseudo-code

---
**Algorithm 1** Regularized Q-learning

---
1: Initialize $\theta_0 \in \mathbb{R}^h$.
2: Set the step-size $(\alpha_k)_{k=0}^{\infty}$, and the behavior policy $\mu$.
3: **for** iteration $k = 0, 1, \ldots$ **do**
4:     Sample $s_k \sim d^\mu$ and $a_k \sim \mu$.
5:     Sample $s_k' \sim P(s_k, a_k, \cdot)$ and $r_{k+1} = r(s_k, a_k, s_k')$.
6:     Update $\theta_k$ using (13).
7: **end for**

---

## B Experiment

### B.1 Experiments

In this section, we present experimental results under well-known environments in Tsitsiklis and Van Roy [1996], Baird [1995], where Q-learning with linear function approximation diverges. In Appendix B.6, we also compare performance under the Mountain Car environment [Sutton and Barto, 2018] where Q-learning performs well. In Appendix B.5, we show experimental results under various step-sizes and $\eta$. We also show trajectories of the O.D.E. of upper and lower comparison systems to illustrate the theoretical results.

### B.2 $\theta \to 2\theta$, Tsitsiklis and Van Roy [1996]

Even when there are only two states, Q-learning with linear function approximation could diverge [Tsitsiklis and Van Roy, 1996]. Depicted in Figure 4a in Appendix B.4, from state one ($\theta$), the transition is deterministic to absorbing state two ($2\theta$), and reward is zero at every time steps. Therefore, the episode length is fixed to be two. Learning rate for Greedy GQ (GGQ) and Coupled Q Learning (CQL), which have two learning rates, are set as $0.05$ and $0.25$, respectively as in Carvalho et al. [2020], Maei et al. [2010]. Since CQL requires normalized feature values, we scaled the feature value with $\frac{1}{2}$ as in Carvalho et al. [2020], and initialized weights as one. We implemented Q-learning with target network [Zhang et al., 2021], which also have two learning rates, without projection for practical reason (Qtarget). We set the learning rate as $0.25$ and $0.05$ respectively, and the weight $\eta$ as two. For RegQ, we set the learning rate as $0.25$, and the weight $\eta$ as two. It is averaged over five runs. In Figure 2a, we can see that RegQ achieves the fastest convergence rate.

### B.3 Baird Seven Star Counter Example, Baird [1995]

Baird [1995] considers an overparameterized example, where Q-learning with linear function approximation diverges. The overall state transition is depicted in Figure 4b given in Appendix B.4.

There are seven states and two actions for each state, which are solid and dash action. The number of features are $h = 15$. At each episode, it is initialized at random state with uniform probability. Solid action leads to seventh state while dashed action makes transition uniformly random to states other than the seventh state. At seventh state, the episode ends with probability $\frac{1}{100}$. The behavior policy selects dashed action with probability $\frac{5}{6}$, and solid action with probability $\frac{1}{6}$. Since CQL in Carvalho et al. [2020] converges under normalized feature values, we scaled the feature matrix with $\frac{1}{\sqrt{5}}$. The weights are set as one except for $\theta_7 = 2$. The learning rates and the weight $\eta$ are set as same as the previous experiment. As in Figure 2b, Our RegQ shows the fastest convergence compared to other convergent algorithms.

## B.4    Diagrams for $\theta \to 2\theta$ and Baird Seven Star Counter Example

The state transition diagrams of $\theta \to 2\theta$ and Baird seven-star example are depicted in Figure 4a and Figure 4b respectively.

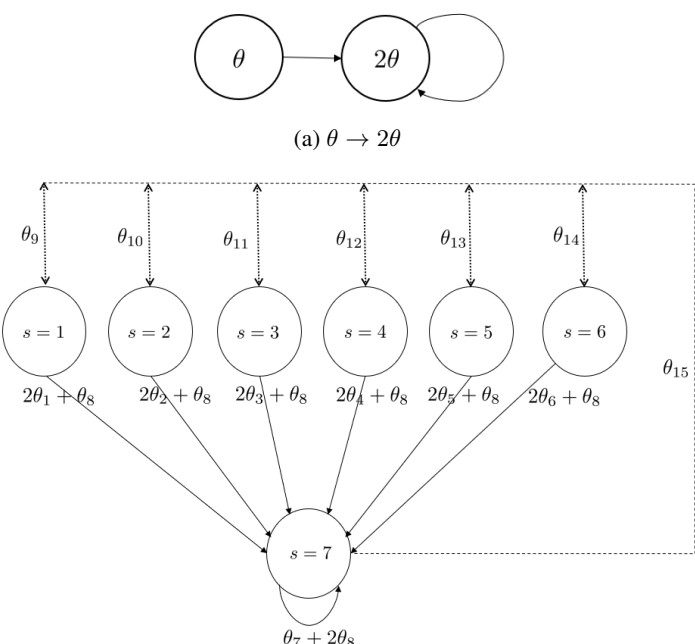

(a) $\theta \to 2\theta$

(b) Baird seven star counter example

Figure 4: Counter-examples where Q-learning with linear function approximation diverges

## B.5    Experiments with varying hyperparameters

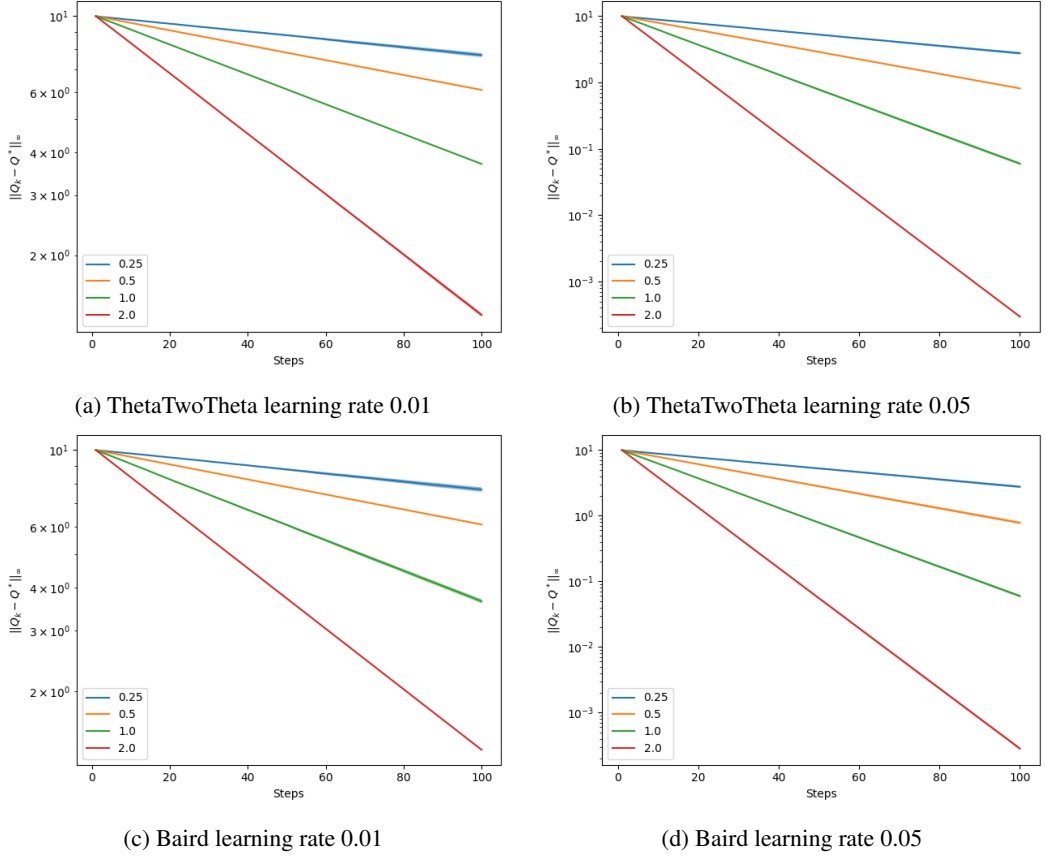

(a) ThetaTwoTheta learning rate 0.01

(b) ThetaTwoTheta learning rate 0.05

(c) Baird learning rate 0.01

(d) Baird learning rate 0.05

Figure 5: Learning curve under different learning rate and regularization coefficient

In Figure 5, we have ran experiments under $\eta \in \{2^{-2}, 2^{-1}, 1, 2\}$, and learning rate $0..01, 0.05$. Overall, we can see that the convergence rate gets faster as $\eta$ increases.

## B.6 Mountain car [Sutton and Barto, 2018] experiment

Mountain Car is environment where state consists of position, and velocity, which are both continuous values. The actions are discrete, accelerating to left, staying neutral, and accelerating to the right. The goal is to reach the top of the mountain quickly as agent gets -1 reward every time step. We use tile-coding [Sutton and Barto, 2018] to discretize the states. We experimented under various tiling numbers and with appropriate $\eta$, it achieves performance as Q-learning does. We ran 1000 episodes for the training process, and the episode reward was averaged for 100 runs during test time. From Table 1, with appropriate $\eta$, RegQ performs comparable to Q-learning.

Table 1: Result of episode reward, step size $= 0.1$. The columns correspond to $\eta$, and rows correspond to number of tiles.

|  | 0 | 0.01 | 0.05 | 0.1 |
|---|---|---|---|---|
| $2 \times 2$ | $-199.993 \pm 0.005$ | $-200.0 \pm 0.0$ | $\mathbf{-199.28 \pm 0.074}$ | $-199.993 \pm 0.005$ |
| $4 \times 4$ | $-196.631 \pm 0.179$ | $\mathbf{-189.903 \pm 0.225}$ | $-194.178 \pm 0.166$ | $-196.631 \pm 0.179$ |
| $8 \times 8$ | $-185.673 \pm 0.305$ | $\mathbf{-163.08 \pm 0.248}$ | $-185.103 \pm 0.219$ | $-185.673 \pm 0.305$ |
| $16 \times 16$ | $-166.893 \pm 0.33$ | $\mathbf{-158.152 \pm 0.251}$ | $-167.934 \pm 0.238$ | $-166.893 \pm 0.33$ |

## B.7 O.D.E. experiment

Let us consider a MDP with $|\mathcal{S}| = 2, |\mathcal{A}| = 2$, and the following parameters:

$$X = \begin{bmatrix} 1 & 0 \\ 0 & 2 \\ 1 & 0 \\ 0 & 2 \end{bmatrix}, \quad D = \begin{bmatrix} \frac{1}{4} & 0 & 0 & 0 \\ 0 & \frac{1}{4} & 0 & 0 \\ 0 & 0 & \frac{1}{4} & 0 \\ 0 & 0 & 0 & \frac{1}{4} \end{bmatrix},$$

$$P = \begin{bmatrix} 0.5 & 0.5 \\ 1 & 0 \\ 0.5 & 0.5 \\ 0.25 & 0.75 \end{bmatrix}, \quad R = \begin{bmatrix} 1 \\ 1 \\ 1 \\ 1 \end{bmatrix}, \quad \gamma = 0.99.$$

For this MDP, we will illustrate trajectories of the upper and lower system. Each state action pair is sampled uniformly random and reward is one for every time step. $\eta = 2.25$ is chosen to satisfy conditions of Theorem 5.1. From Figure 6, we can see that the trajectory of the original system is bounded by the trajectories of lower and upper system.

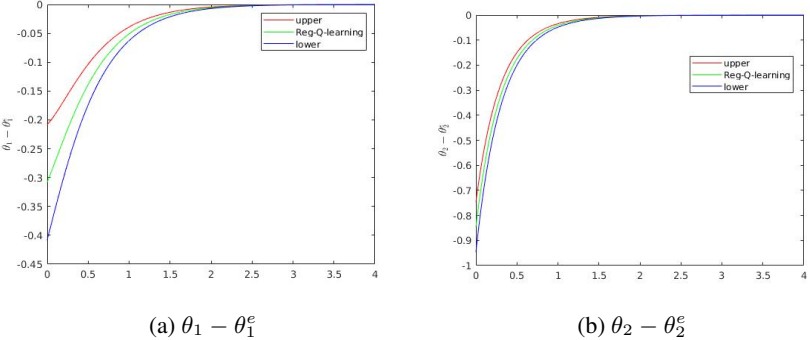

(a) $\theta_1 - \theta_1^e$        (b) $\theta_2 - \theta_2^e$

Figure 6: O.D.E. results

