# OpenReview forum: "Regularized Q-Learning"
_NeurIPS.cc/2024/Conference — NeurIPS 2024 poster_

### Official Review · Reviewer_x1RW · 2024-07-10

**Soundness:** 1
**Presentation:** 1
**Contribution:** 3
**Rating:** 5
**Confidence:** 3

**Summary:**

This paper provides Q-learning convergence (asymptotic) in linear architectures with regularization power. Their algorithm is tested on mountain car example.

**Strengths:**

Q-learning convergence in linear architectures is an important problem in RL.

**Weaknesses:**

-  The analysis follows ODE style analysis from Borkar and Meyn. So it is asymptotic. However, non-asymptotic guarantees (rate of convergence) can be provided when assuming non-zero stationary distribution (like Assumption 2.1). For e.g. see Chen et al, 22. There are research on extension to QL like TD-Learning convergence in linear architectures. So I think with the current tools one can give such finite time guarantees with some more effort. This work relies on old works such as Gosavi [2006] and Melo et al., [2008]. *Although, I would be curious to know where the current hurdles are.*
  - A recent work titled "Regularized Q-Learning with Linear Function Approximation" (https://arxiv.org/pdf/2401.15196) are providing non-asymptotic results for a similar problem. I have not looked at the details, but since this appeared in arxiv in Jan 2024, please do include what the contributions are of this work compared to this one.

- Section 3.2 can be further improved w.r.t writing. I believe Lemma 3.3, 3.4, 3.5 are provided as independent results to satisfy Eq.(11) (which is crucial for existence and uniqueness of RPBE solution). But I am curious to know why 3 different Lemmas are provided.
  - How is eq.(15) constructed for $\eta$? I understand Lemma 3.3 helps for (S1). But I am not sure how (S2) came about.
  - It is mentioned at line 228 that $\eta>2$ is enough for Lemma 3.3. But $\eta<1$ for Lemma 3.4. I really hope Lemma 3.3 and 3.4 are **not required** to be satisfied simultaneously for the current results to hold.
  - Where is Lemma 3.1 used? We are in linear architecture setting. So Lemma 3.1 (b) inequality which involves $S\times A$ must be avoided. Linear architectures are helpful only when one can replace rate of convergences involving $S\times A$ can be replaced by feature-size $h\ll SA$.

My score reflects this review provided here.

**Questions:**

na

---

> ### Author Rebuttal · Authors · 2024-08-06
>
> **Q1** *The analysis follows ODE style analysis from Borkar and Meyn. So it is asymptotic. However, non-asymptotic guarantees (rate of convergence) can be provided when assuming non-zero stationary distribution (like Assumption 2.1). For e.g. see Chen et al, 22. There are research on extension to QL like TD-Learning convergence in linear architectures. So I think with the current tools one can give such finite time guarantees with some more effort. This work relies on old works such as Gosavi [2006] and Melo et al., [2008]. Although, I would be curious to know where the current hurdles are*
>
> **A1** Thank you for the valuable comments. As the reviewer mentioned, we believe finite-time analysis of RegQ is possible. However, our primal goal is to develop a new Q-learning algorithm with linear function approximation that converges under relaxed or different scenarios. We consider proving the asymptotic convergence of Q-learning with linear function approximation as an initial step and the finite-time bound can be also proved following the spirit of related literature. We appreciate the provision of interesting research topics, which remain for future exploration. According to the reviewer's comment, the related discussions will be added in the revision.
>
> Moreover, we would like to note that our work does not rely on the work of Melo et al.,[2008]. We also note that Melo et al.,[2008] requires strong assumptions on the behavior policy to be met, whereas we do not requires such assumptions. Furthermore, we provided a comparison with existing algorithms in G1 of the global response, which clarifies current hurdles and the novelty of our approach. Following the reviewer's comment, we will add the associated discussions in the revised manuscript.
>
> **Q2** *A recent work titled "Regularized Q-Learning with Linear Function Approximation" are providing non-asymptotic results for a similar problem. I have not looked at the details, but since this appeared in arxiv in Jan 2024, please do include what the contributions are of this work compared to this one.*
>
>
> **A2** Thank you for the insightful comment. The regularization considered in ``Regularized Q-Learning with Linear Function Approximation'' requires to be bounded, e.g., an entropy regularizer. The $l$-2 type regularization which we consider in our work does not fall into this category, and the extension is non-trivial. Following the reviewer's recommendation, we will incorporate the discussions on the comparisons in the revised manuscript.
>
>
> **Q3** *Section 3.2 can be further improved w.r.t writing. I believe Lemma 3.3, 3.4, 3.5 are provided as independent results to satisfy Eq.(11) (which is crucial for existence and uniqueness of RPBE solution). But I am curious to know why 3 different Lemmas are provided.*
>
>
> **A3** Thank you for helping us to improve the clarity of our manuscript. We provided three different lemmas because each addresses different scenarios for (11) to hold. Lemma 3.3 covers when $\eta$ is larger than certain threshold, Lemma 3.4 considers the case when $\eta$ is nearby the origin, and Lemma 3.5 applies for for all $\eta$. Following the reviewer's comment, we will improve the clarity in the revised manuscript.
>
>
> **Q4** *How is eq.(15) constructed for $\eta$? I understand Lemma 3.3 helps for (S1). But I am not sure how (S2) came about.*
>
> **A4** We thank the reviewer for the insightful comments. The conditions in (S1) and (S2)  is used to guarantee $(A_{\pi_{X\theta_k}} + \eta I)$ to have strictly negatively dominant diagonal or to be a negative definite matrix, respectively. In particular (S2) comes from the item 2 in Lemma 2.5, which guarantees the asymptotic stability of a switched system. The proof for the derivation of (S2) is given in Lemma A.6. in the Appendix of the manuscript. Note that the conditions (S1) and (S2) do not necessarily imply each others, which are discussed in Appendix A.15 of the manuscript. We will add the above discussion in the revised manuscript.
>
>
> **Q5** *It is mentioned at line 228 that $\eta>2$ is enough for Lemma 3.3. But $\eta<1$ for Lemma 3.4. I really hope Lemma 3.3 and 3.4 are not required to be satisfied simultaneously for the current results to hold.*
>
>
> **A5** Thank you for the valuable comments. Lemma 3.3 and Lemma 3.4 are only sufficient conditions but not necessary conditions for (11) to hold. Therefore, the two statements do not contradict each other. Moreover, we would like to note that Lemma~3.4 does not require the condition $\eta<1$. It only assumes the scenario $|| \gamma \Gamma ||_\infty < 1$. Following the reviewer's comment, we will clarify these points in the revised manuscript.
>
> **Q6** *Where is Lemma 3.1 used? We are in linear architecture setting. So Lemma 3.1 (b) inequality which involves $S\times A$ must be avoided. Linear architectures are helpful only when one can replace rate of convergences involving $S\times A$ can be replaced by feature-size $h<<SA$*
>
> **A6** We thank the reviewer for the insightful comments. Lemma 3.1 is provided for a theoretical understanding on how $\Gamma_{\eta}$ behaves as a function of $\eta$.
>
> The inequality in Lemma 3.1 (b) demonstrates that $\Gamma_{\eta}$ remains bounded for all $\eta$ and does not diverge. We do not directly use the upper bound in Lemma 3.1 for our analysis, and it is not related to the rate of convergence. We will clarify this in the revised manuscript.

---

> > ### Comment · Reviewer_x1RW · 2024-08-13
> >
> > Thank you for reflecting on the reviews. I will keep my score as the updates for Lemma 3.1-3.5 require some non-trivial updates that need to be evaluated further, which will be out of scope in this rebuttal period. Good luck.

---

> ### Author Response · Authors · 2024-08-13
>
> We thank the reviewer for the response in the discussion period. However, we kindly disagree that Lemma 3.1-3.5 needs further evaluation or  poses logical errors.
>
> 1) Lemma 3.1 :  As mentioned in Q6, the lemma is only used to show that $\Gamma_{\eta}$ is bounded for any $\eta$, and we do not use the upper bound which is dependent on $|S||A|$ in other result or proof.
>
> 2) Lemma 3.3-3.5 : Regarding Q5 in the initial rebuttal, the two lemmas do not contradict each other because they are only sufficient conditions for (11) to hold. They are totally independent results. Each lemmas cover different scenarios, and hence there are no logical errors between them. Furthermore, Lemma 3.3 covers the most practical scenario. If we scale the feature matrix such that $\max(||X||\_{\infty},||X||\_{\infty})<1$, then choosing $\eta>2$ satisfies the condition (11) by Lemma 3.3 in the manuscript. Scaling the values of feature matrix is a commonly employed technique in the both theoretical literature or in practice.
>
> We again thank the reviewer for the engagement in the discussion, and kindly request the re-evaluation of our manuscript.

---

### Official Review · Reviewer_cMMt · 2024-07-12

**Soundness:** 3
**Presentation:** 3
**Contribution:** 3
**Rating:** 4
**Confidence:** 3

**Summary:**

This paper introduces a novel approach, RegQ, which is a framework for dealing with linear approximation of Q-function. Compared to the instability of the traditional Q-learning with function approximator, which is known as the deadly triad, RegQ addresses this problem by regularization term, making the algorithm more stable. Also the theoretical approach ensures convergence of linear function approximation.

**Strengths:**

This paper gives a novel approach with theoretical rigor.
The most noteworthy strength of this paper is that it tackles a practical problem: the instability of Q-learning with linear function approximator, compared to other theoretical paper that gives impractical solutions.
Although the theory is highly technical, this paper gives a good logical explanation, also Figure 1 helps readers to understand the proposed projection operator.

**Weaknesses:**

Most of the concerns arose from the lack of experiment scope, prior work comparison, and implementation.
Authors claim about the strengths of the RegQ algorithms, but lots of claims are not confirmed by experiments.
Also, the claims are given with the comparison of the prior work, but experimental results are not sufficient to validate those claims.
It might be helpful to gain confidence about the novelty of the paper,
Also, it is hard to get a precise understanding of the algorithm since there are lack of implementation details.

**Questions:**

In equation (13), what delta_k means?
Also, in line 290, why m_k+1 is an i.i.d. noise? Is it an assumption, or can it be proved?

In Figure2, the y-axis means the max norm of Q-values. I’m wondering about the value of y axis. At the initial stage of the episode, all of the algorithms have already shown 1e-18, and 1e-20 values, and what the authors claim is RegQ converges when others fail. My question is this: why initial point or other algorithms( which is shown 1e-18 error) is not considered as converged? Also, since values are
too small, how could authors prevent the contamination by the floating point error? It will be easy to understand if the authors give more explanation about experiments.

In Figure 1c, I can’t find the details about the phases: ‘Blowing up phase’, and ‘Shrinking phase’.
I read the author’s error analysis, but there’s no explanation about the phase. Is it just a intuitive naming on the convergence behavior? If not, could you elaborate more about it in the theoretical context?

---

> ### Author Rebuttal · Authors · 2024-08-06
>
> **Q1.** *Most of the concerns arose from the lack of experiment scope, prior work comparison, and implementation. Authors claim about the strengths of the RegQ algorithms, but lots of claims are not confirmed by experiments. Also, the claims are given with the comparison of the prior work, but experimental results are not sufficient to validate those claims. It might be helpful to gain confidence about the novelty of the paper. Also, it is hard to get a precise understanding of the algorithm since there are lack of implementation details.*
>
> **A1.** Thank you for the constructive comments. We would like to first note that due to the page limits, all the experiments are included in the Appendix B.. Similarly, the implementation details are given in the Appendix B. Moreover, we want to clarify that our main focus is on the theoretical analysis of proposed RegQ algorithm and regularized projected Bellman equation. We proposed a **convergent** algorithm under linear function approximation and mild assumptions and provided a thorough analysis on the conditions for the convergence. The comparison with prior works are mainly in theoretical sense, e.g., weaker assumption on the behavior policy or relaxing additional assumptions used in prior works. Moreover, the analysis on the properties of regularized projected Bellman equation is new in the literature.
>
> The experiments are provided to verify the convergence of the proposed algorithm and provide further insights on its behavior, e.g., convergence rate. Our claim is that RegQ can show faster convergence rate than other algorithms under certain scenarios, which is verified in Section 6. An intuitive reason why our algorithms shows faster convergence is that the baseline algorithms ( CQL, targetQ, and GreedyGQ ) are basically a two-time scale algorithm whereas our proposed algorithm uses a single-time scale step-size. Moreover, regarding the implementation detail, a pseudo-code is given in Appendix A.16. Following the reviewer's comments, we have added the discussion in the revised manuscript.
>
>
> **Q2.** *In equation (13), what $\delta_k$ means? Also, in line 290, why $m_{k+1}$ is an i.i.d. noise? Is it an assumption, or can it be proved?*
>
>
> **A2.** We thank the reviewer for the constructive comments. For the first question, $\delta_k$ is the TD error, which is defined below in the equation (2) in the manuscript. $m_{k+1}$ is not an i.i.d. noise but the Martingale difference sequence, which includes the i.i.d. noise scenario as a special case, and which is shown in Lemma A.14 in the Appendix. We will correct the typo in the revision. We would like to note that the Martingale difference sequence scenario is an assumption, and it cannot cover the case that the transitions are sampled from a single trajectory (or Markovian sampling scenario).
> In [1], the authors demonstrated an extension of the Borkar and Meyn Theorem to the Markovian sample cases. Since our proof relies on Borkar and Meyn Theorem, our result can also be extended to establish convergence under Markovian sample cases. We have incorporated this result into the revised manuscript.
>
> **Q3.** *In Figure2, the y-axis means the max norm of Q-values. I’m wondering about the value of y axis. At the initial stage of the episode, all of the algorithms have already shown 1e-18, and 1e-20 values, and what the authors claim is RegQ converges when others fail. My question is this: why initial point or other algorithms( which is shown 1e-18 error) is not considered as converged? Also, since values are too small, how could authors prevent the contamination by the floating point error? It will be easy to understand if the authors give more explanation about experiments.*
>
> **A3.** Thank you for the valuable comments. We agree with the reviewer. For clarification on the error bound, we have provided the first 50 steps in Figure 1a and 2a in the pdf file attached to the global response. The confusion arose because in the original manuscript,  we have plotted the x-axis by episode, which consists of number of updates. In the corrected plot, we plotted the x-axis with number of updates. The initial point is not the convergence point because the error is larger than one, as can be seen in the corrected plot. According to the reviewer's comment, the figures have been replaced with the new ones in the revision, and more details of the figures have been newly added.
>
> Moreover, we would like to clarify that our claim does not mean that other algorithms fail while RegQ converges. Our claim is that under certain situations, the convergence rate of RegQ can be faster than that of other algorithms. An intuitive reason why our algorithms shows faster convergence is that the baseline algorithms ( CQL, targetQ, and GreedyGQ ) are basically a two-time scale algorithm whereas our proposed algorithm uses a single-time step-size. Following the reviewer's comment, we will update our manuscript.
>
> **Q4.** *In Figure 1c, I can’t find the details about the phases: ‘Blowing up phase’, and ‘Shrinking phase’. I read the author’s error analysis, but there’s no explanation about the phase. Is it just a intuitive naming on the convergence behavior? If not, could you elaborate more about it in the theoretical context?*
>
> **A4.** Thank you helping us to improve the clarity of our manuscript. Regarding the terms shrinking and blowing up phases, we have provided a response G2 in the global rebuttal. To clarify this further, the related discussions have been newly added in the revision.
>
> **References**
>
> [1] Liu, Shuze, Shuhang Chen, and Shangtong Zhang. "The ODE Method for Stochastic Approximation and Reinforcement Learning with Markovian Noise." arXiv preprint arXiv:2401.07844 (2024).

---

> ### Author Response · Authors · 2024-08-13
>
> We appreciate the reviewer's time and effort in reviewing our manuscript. If there are any additional concerns, please let us know since the discussion period is coming to its end. Otherwise, we kindly request a re-evaluation of our manuscript.

---

### Official Review · Reviewer_mviV · 2024-07-12

**Soundness:** 3
**Presentation:** 3
**Contribution:** 3
**Rating:** 5
**Confidence:** 3

**Summary:**

This paper proposes a new Q-learning variant with linear function approximation called RegQ, and proves that its ODE form converges (even when associating it with linear function approximations).

Q-learning is famously known to be affected by the 'deadly triad': it tends to diverge in practice when combined to off-policyness, function approximation and bootsrapping. Analyzing formally the convergence of Q-learning combined with simple linear function approximations helps understanding more precisely the mechanisms at play in the deadly triad, and paves the way toward RL with more solid foundations.

Authors show empirically that RegQ is faster than two related algorithms with guaranteed convergence. Besides, RegQ relies on a single time-scale, while the other baseline algorithms use two time-scales.

**Strengths:**

- One of the most appealling properties of the framework proposed in this paper is its simplicity: just regularizing projected Bellman equations allows to obtain convergence proofs without relying on several artificial assumptions.

**Weaknesses:**

- In the related work, the paper lists several existing works proving the convergence of Q-learning under linear function approximation with some theoretical assumptions.
The assumptions made in the present paper are weaker than most of the ones of existing works, which include restrictions on the Markov chain types, dependency between behavior and target policy, other guarantees than convergence, etc. However, the closest work is by Lee and He (2019), and the present paper does not explicitely state in what way the assumptions are now weaker than in Lee and He. It states that Lee and He's assumption on the behavior and feature matrix seems too stringent to check in practice, but nothing more precise. When Lee and He improved the sufficient condition of Melo et al. (2008), they showed that their new condition was strictly weaker than the previous one, and I believe that a similar analysis should be made. More structure in the comparison would make the paper look less like an incremental modification of Lee and He (2019).

- The presented work follows the direction proposed in Lee and He (2019) by reducing the convergence analysis to that of a switching system, establishing simpler upper and lower bound systems and applying the Borkar and Meyn theorem to obtain a proof of the asymptotic convergence. There are of course différences in the approach, but they are in the details.

- It can be regretted that the most interesting and novel parts of the paper are in Appendix, which shows that the conference format. might not be the best fit for it.

**Questions:**

- Although there are empirical evaluations, including one on Mountain Car, more ambitious empirical tests with regularized variants of Q-learning-based deep RL algorithms would be interesting.

- Could the proposed algorithm fit the framework of regularized MDPs introduced in ["A Theory of Regularized Markov Decision Processes", Geist et al., 2019]? If yes, could results in ["A Theory of Regularized Markov Decision Processes", Geist et al., 2019] be directly applied to the proposed method, including for instance the analysis of the changes on the optimal policy due to regularization (error bound)? Including a discussion on this existing framework in the paper seems relevant.

Typos:

l9: 'has known to diverge' -> 'was known to diverge'

l64: "guarantees convergence" => "guarantee convergence"

**Limitations:**

Limitations are well addressed.

---

> ### Author Rebuttal · Authors · 2024-08-06
>
> **Q1** *In the related work, the paper lists several existing works proving the convergence of Q-learning under linear function approximation with some theoretical assumptions. The assumptions made in the present paper are weaker than most of the ones of existing works, which include restrictions on the Markov chain types, dependency between behavior and target policy, other guarantees than convergence, etc. However, the closest work is by Lee and He (2019), and the present paper does not explicitely state in what way the assumptions are now weaker than in Lee and He. It states that Lee and He's assumption on the behavior and feature matrix seems too stringent to check in practice, but nothing more precise. When Lee and He improved the sufficient condition of Melo et al. (2008), they showed that their new condition was strictly weaker than the previous one, and I believe that a similar analysis should be made. More structure in the comparison would make the paper look less like an incremental modification of Lee and He (2019).*
>
> **A1.** We thank the reviewer for the valuable insights. Lee and He (2919) considered the following condition:
> $$\phi^\top_i D + \phi^\top_i \gamma DP \Pi_{\pi}\sum_{j\in \{1,2,\dots,n\}}\phi_j<0 ,\quad \pi \in \Theta_{\Phi},$$
> where $\Theta_{\Phi}:= \{ \pi\in\Theta : \pi(s)=arg\max_{a\in\mathcal{A}} (\Phi\theta)(s,a) ,\forall s \in \mathcal{S},\theta\in\mathbb{R}^n \}$, $\Theta$ is the set of greedy policies, and
> \begin{align*}
>  \phi_i= \begin{bmatrix}
>     \phi_i(1,1) & \phi_i(2,1) & \cdots & \phi_i(|\mathcal{S}|,|\mathcal{A}|)]^{\top} \end{bmatrix}.
> \end{align*}
>  The above condition is strict because it needs to hold for all the policy $\pi\in\Theta_{\Phi}$. In contrast, we do not require such condition. We only require the regularization coefficient $\eta$ to be larger than certain value, which can be easily met by scaling the feature matrix. In particular, the condition on $\eta$ can be met by scaling the norm of feature matrix to be smaller than one, and $\eta$ only needs to be larger than two. This follows from Lemma 3.3 in the manuscript. Moreover, we note that feature scaling is widely used in practice. Following the reviewer's comments, we will incorporate the discussion in the revised manuscript.
>
>
> **Q2** *The presented work follows the direction proposed in Lee and He (2019) by reducing the convergence analysis to that of a switching system, establishing simpler upper and lower bound systems and applying the Borkar and Meyn theorem to obtain a proof of the asymptotic convergence. There are of course differences in the approach, but they are in the details.*
>
>
> **A2** We appreciate the reviewer's constructive feedback. While our approach aligns with the principles outlined by Lee and He (2019), the following points are new in the literature:
>
> 1) We established the theoretical conditions for regularization term to ensure convergence of the algorithm; Moreover, as mentioned in A1, simply adding a regularization term allows to weaken the assumptions used in prior works including Lee and He (2019);
>
> 2) We characterized the existence and uniqueness of the regularized projected Bellman equation depending on $\eta$;
>
> 3) A tight error bound between the solution regularized projected Bellman equation and true solution, $Q^*$ is provided.
>
> Each of the above points has not been studied previously, and our work provided a thorough analysis for the above points . Following the reviewer's comments, we will clarify this in the revised manuscript.
>
>
> **Q3** *It can be regretted that the most interesting and novel parts of the paper are in Appendix, which shows that the conference format. might not be the best fit for it.*
>
> **A3** We thank the reviewer for the valuable comments. As the reviewer mentioned, the proofs are deferred to the Appendix due to the space limit. However, we highlight that our contribution lies is the analysis of the properties of RPBE and the development of RegQ. We have provided thorough analysis of RPBE in Section 3, and have further elaborated on this, including additional details on Figure 1c of the manuscript in G1 of the global rebuttal. Additionally, as answered in A1 and A2, we will clarify the points regarding the analysis of RegQ. Following the reviewer's comments, we have incorporate these clarifications in the revised manuscript.
>
> **Q4** *Although there are empirical evaluations, including one on Mountain Car, more ambitious empirical tests with regularized variants of Q-learning-based deep RL algorithms would be interesting.*
>
> **A4** We thank the reviewer for the insightful comments. In the revised manuscript, we will consider more ambitious empirical evaluations on the variants of Q-learning-based deep RL algorithms.
>
> **Q5** *Could the proposed algorithm fit the framework of regularized MDPs introduced in ["A Theory of Regularized Markov Decision Processes", Geist et al., 2019]? If yes, could results in ["A Theory of Regularized Markov Decision Processes", Geist et al., 2019] be directly applied to the proposed method, including for instance the analysis of the changes on the optimal policy due to regularization (error bound)? Including a discussion on this existing framework in the paper seems relevant.*
>
> **A5** Thank you for the insightful comments. The domain of the regularization term used in Geist et al., 2019 is restricted to a probability simplex, for example an entropy regularization. The work focuses on the regularization over the policy space instead of the Q-function, and also focuses on the policy iteration algorithms instead of reinforcement learning or Q-learning. The $l$-2 type regularization we consider does not fall into this category, and the extension is non-trivial. We will incorporate the discussion in the revised manuscript.

---

> > ### Comment · Reviewer_mviV · 2024-08-12
> >
> > Thank you for these answers. As I still consider the work an incremental modification of Lee and He 2019, I keep my recommendation of borderline acceptance.

---

> ### Author Response · Authors · 2024-08-13
>
> **Q1.** *As I still consider the work an incremental modification of Lee and He 2019*
>
> **A1** We thank the reviewer for the valuable feedback. We want to note that our contribution is not only proving the convergence of RegQ but also lies in thorough investigation of theoretical properties of regularized projected Bellman equation (RPBE), which has been explained in A2 in the initial response. We have provided thorough theoretical investigation of existence, uniqueness, and quality of the solution of RPBE. This is a unique contribution of our work and has not been presented in Lee and He 2019. Following the reviewer's comment, we will add the discussion in the revised manuscript.
>
> We thank the reviewer for the engagement in the discussion period, and providing constructive comments to improve the quality of the paper.

---

### Official Review · Reviewer_yA4E · 2024-07-12

**Soundness:** 4
**Presentation:** 3
**Contribution:** 3
**Rating:** 8
**Confidence:** 3

**Summary:**

The paper introduces a new regularized Q-learning algorithm "RegQ" suitable for linear function approximation, which essentially adds an $\ell^2$ regularization term to the TD error in semi-gradient Q-learning. The authors prove that this addition ensures convergence of the algorithm and analyze the error with respect to the unregularized solution.

**Strengths:**

The paper centers on an important issue in reinforcement learning: the deadly triad, which is the failure of off-policy TD algorithms when combined with function approximation. This issue has been addressed in practical (deep) RL by expensive methods such as target networks. This paper proposes a simpler solution (specifically for the case of linear function approximation): regularization of the TD error. The paper is very clearly written and provides extensive sections on the background and related work. Their assumptions seem reasonable, and their analysis is rigorous. The overall contribution is highly significant.

**Weaknesses:**

1. The biggest weakness of this paper is the limited experiment section. I understand that this is a theoretical work, and of course don't expect any large-scale experiments. However, the two experiments they do list are not properly explained. (What are these baseline algorithms?) The takeaway from the experiments is that their method has a faster convergence rate, but it is not explained why. The convergence rate is not mentioned elsewhere in the paper, where the focus lies on proving convergence, where other methods do not converge! It would be great to show an environment where RegQ converges while the baseline methods do not, or where the error of RegQ's approximate solution $\theta_\eta^\star$ is smaller than the baselines'.
2. Please comment on Assumption 2.2 (orthogonality of columns of $X$). Could it be relaxed? In high-dimensional ($|\mathcal S||\mathcal A|$) spaces, $h \ll |\mathcal S||\mathcal A|$ _random_ vectors are nearly orthogonal with high probability. Could this be used to show that the result will hold with high probability when using random features?
3. Figure 1c is not properly explained. What are the "shrinking" and "blowing up" phases? Why does the vector $x$, located on the _unit circle_ have norm $0$?
4. For Lemma 3.5, it is assumed that $X^\top D X = aI$. I understand that this is just an example of when a solution to the RPBE exists, but it should be clarified that this assumption is very unrealistic (earlier you wrote that $h \ll |\mathcal S||\mathcal A|$, which is a contradiction).
5. In line 245, you state that, as $\theta^\star = \theta_\eta^\star$ if $\eta = 0$, it holds that $\theta^\star \to \theta_\eta^\star$ if $\eta \to 0$. You are implicitly assuming that $\theta_\eta^\star$ is a continuous function of $\eta$ (at $\eta = 0$), which you should at least mention, if not prove.
6. (minor) In the introduction, you do not mention at all that you also analyze the error $\theta_\eta^\star - \theta^\star$, which to me is a very important part of your work, and would not be out of place in the "summary of main contributions" at the end of your introduction.
7. (minor) As you talk about the deadly triad and how it has been addressed practically in deep RL, you might also want to cite "Deep reinforcement learning and the deadly triad" by van Hasselt et al. (2018).
8. (typo) In line 215, you swap (9) and (4), changing the meaning in a significant way.

**Questions:**

1. Where does equation (6) come from? To solve an equation like $A\theta = b$ iteratively, I would construct a loss function $L(\theta) = \frac{1}{2}||A\theta - b||^2$ and do gradient descent: $\theta_{k+1} = \theta_k - \alpha_k\nabla L(\theta_k)$, where $\nabla L(\theta_k) = A^\top(A\theta - b) + E$, with $E$ containing additional terms if $\nabla A \neq 0$. Equation (6) looks similar at first but is in fact quite different. Could you explain how you arrived at equation (6)?
2. How should equation (15) be interpreted? How "large" does $\eta$ actually have to be to ensure convergence?

**Limitations:**

The authors adequately address the limitations of their work.

---

> ### Author Rebuttal · Authors · 2024-08-06
>
> We thank the reviewer for the thoughtful feedback, and the time and effort for reviewing our paper. Following the reviewer's comments, we have added the related discussion in the revised manuscript:
>
> **Q1** *The biggest weakness of this paper is the limited experiment  ... . The two experiments they list are not properly explained. (What are these baseline algorithms?) The takeaway from the experiments is that their method has a faster convergence rate, but it is not explained why. ... It would be great to show an environment where RegQ converges while the baseline methods do not, or where the error of RegQ's approximate solution is smaller than the baselines.*
>
> **A1.** Thank you for the insightful comments. The baseline algorithms are Q-learning variants that converges under linear function approximation, which are explained in Appendix Section D. The algorithms are Coupled Q-learning (CQL), GreedyGQ, and algorithms with target-network update (targetQ). As the reviewer suggested, we have newly provided an example, where RegQ converges while one of the baseline algorithm, CQL, does not converge in the PDF file attached in the global response. This is because CQL requires the norm of the feature matrix to be smaller than one. Meanwhile, we could not verify practical examples showing divergence of GreedyGQ or targetQ while RegQ diverges or vice-versa. This is because the Q-learning variants are developed to guarantee convergence to some solutions. However, we summarized the difference on the theoretical points in G1 in the global response.
>
> Lastly, the experiment results show that RegQ has a faster convergence rate under some scenarios. An intuitive reason for this is because the baseline algorithms are basically a two-time scale algorithms whereas RegQ uses a single-time step-size.  A detailed exploration of convergence rates is a promising direction for future research. Following the reviewer's comments, we have added the related discussions in the revised manuscript.
>
> **Q2** *Please comment on Assumption 2.2 (orthogonality of columns of $\Phi$). Could it be relaxed?*
>
> **A2** Thank you for the insightful comments on the our paper. The assumption is required for the construction of comparison systems in the switched system analysis, and it seems that relaxing this condition in the current phase is non-trivial. However, the orthogonality assumption is not very restrictive and can be met easily in practice. For instance, we can use orthogonal Fourier basis functions as feature functions. Additionally, it may be possible to develop more advanced techniques in the future that can relax this assumption through coordinate transformations. This could be an interesting agenda for future research. Moreover, as the reviewer suggested, initialization of random vectors can guarantee such condition with high probability. We believe the suggested arguments can be used to justify the assumption.
>
> **Q3** *In Figure 1c, what are the "shrinking" and "blowing up" phases? Why does the vector $x$, located on the unit circle have norm $0$?*
>
> **A3** Thank you for the careful investigation of our manuscript. This a typo and the correct expression is $||x||_{\infty}=1$. We will correct this in the revised manuscript. Regarding the terms shrinking and blowing up phases, we have provided a response G2 in the global rebuttal.
>
> **Q4**  *For Lemma 3.5, it is assumed that $X^\top D X = aI$. .. just an example ... but it should be clarified that this assumption is very unrealistic (earlier you wrote that  $h\ll  |{\cal S}||{\cal A}|$, which is a contradiction).*
>
> **A4** Thank you for pointing out the issue. Following the reviewer's comments, we will clarify this point in the manuscript that such cases are unrealistic and differ from the case of $h\ll  |{\cal S}||{\cal A}|$.
>
>  **Q5** *In line 245, you state that, as* $\theta^*=\theta_{\eta}^*$ *if $\eta=0$, it holds that* $\theta^*\to \theta^*_{\eta}$ *if $\eta\to 0$. You are implicitly assuming that is a continuous function of $\eta$ at 0.*
>
> **A5**  We have provided a proof in G3 in the global response. We would like to note that before guaranteeing the continuity of $\theta^*_{\eta}$, it should at least exist in a neighborhood of $\eta = 0$. Therefore, to guarantee the existence of $\theta^*_{\eta}$ around $\eta = 0$, we have added the condition $\gamma||\Gamma||_\infty<1$. Then, under the existence, we can prove the continuity of $\theta^*_{\eta}$ at $\eta=0$. Moreover, this result can be easily extended to the case that the solution $\theta^*_{\eta}$ exists for a positive $\eta > 0$.
>
>  **Q6** *Where does equation (6) come from?*
>
> **A6**  Equation (6) is not a gradient of any objective function. It iteratively solves an equation $A\theta=b$ by $\theta_{k+1}\leftarrow \theta_k + \alpha(A\theta_{k}-b)$., which is a widely used algorithm called a Richardson iteration [1]. It simply updates by the difference of the left and right term of the equation.
>
> **Q7** *How should equation (15) be interpreted? How "large" does $\eta$ actually have to be to ensure convergence?*
>
> **A7**  In equation (15), the terms (S1) and (S2) correspond to the condition for the stability of the switched system in each item of Lemma 2.5 to be satisfied, respectively. As noted in Appendix A.15, each condition covers different scenarios. However, the bound can be always chosen to be small because it only needs to be larger than the minimum of two quantity, (S1) and (S2). As show in Lemma 3.3, $\eta>2$ only needs to be met if we use feature scaling, which is widely used in practice. However, we note that there are several cases when (S2) can be smaller than (S1), as noted in the Appendix A.15.
>
> We thank again the reviewer for the constructive comments. Following the reviewer's comments, we have incorporated the above discussions in the revised manuscript.
>
> **References**
> [1] Kelley, Carl T. Iterative methods for linear and nonlinear equations. Society for Industrial and Applied Mathematics, 1995.

---

> > ### Comment · Reviewer_yA4E · 2024-08-12
> >
> > Thank you for your detailed response. I will keep my score as is.

---

### Official Review · Reviewer_haoJ · 2024-07-12

**Soundness:** 3
**Presentation:** 3
**Contribution:** 2
**Rating:** 5
**Confidence:** 4

**Summary:**

Q-learning is a popular RL algorithm. With function approximation, though, it is known that this algorithm can diverge. This issue is attributed to the `deadly triad': off-policy learning, bootstrapping, and function approximation. This work addresses this issue in the context of linear function approximation. Specifically, this work proposes a novel called Regularized Q-learning, in which a suitable regularization term is added to the standard update rule. The key result (Theorem 5.2) is that this modified algorithm almost surely converges. The proof is based on the switching systems.

**Strengths:**

S1. The paper introduces a new Q-learning algorithm called Regularized Q-learning (RegQ), which ensures convergence under linear function approximation. This addresses the known instability issue in traditional Q-learning with function approximation.

S2. The paper uses the switched system theory to derive RegQ's convergence.

**Weaknesses:**

W1. The present work studies **only** the case of Q-learning with linear function approximation under a **fixed behavior policy**. This approach is extremely restrictive and practically not very useful since the quality of the resulting policy critically depends on the choice of the behavior policy. Specifically, as stated by Melo, Meyn, and Ribeiro (2008), for the approximate Q-learning algorithm to discover the optimal policy, the behavior policy would need to be close to the optimal policy itself, which is not feasible. For other choices of behavior policy, the policy estimated by the algorithm could be significantly different from the optimal policy. While Lemma 3.7 provides some guarantees, it is unclear how this result relates the greedy policy, with respect to $X \theta_{\eta}^*,$ to the optimal policy.

This is why $\epsilon$-greedy exploration is commonly used in practice. However, the current work does not address this important case. Under $\epsilon$-greedy exploration, several recent studies [1] -- [3] have shown that Q-learning with function approximation suffers from various significant issues beyond instability. Notably, this algorithm can converge to non-locally optimal policies, sometimes even the worst, and exhibit policy oscillation. It remains unclear if the regularization term proposed in the present work would effectively address these issues with $\epsilon$-greedy exploration.

References:

[1] Patterson, A., Neumann, S., White, M. and White, A., 2023. Empirical design in reinforcement learning. arXiv preprint arXiv:2304.01315.

[2] Young, K. and Sutton, R.S., 2020. Understanding the pathologies of approximate policy evaluation when combined with greedification in reinforcement learning. arXiv preprint arXiv:2010.15268.

[3] Gopalan, A. and Thoppe, G., 2022. Demystifying Approximate Value-based RL with $\epsilon $-greedy Exploration: A Differential Inclusion View. arXiv preprint arXiv:2205.13617.

**Questions:**

L1. The current work only studies the setting where the $(s_k, a_k, r_{k + 1}, s_{k + 1})_{k \geq 0}$ sequence is sampled in an IID fashion in each iteration. Do you think the results carry over to the scenario with Markovian samples?

L2. Are there any realistic examples of the feature matrix $X$ for which the condition in (11) is guaranteed?

Minor issues:

M1. Line 90: Shouldn't you emphasize what s_0's distribution is?

M2. Line 96: Which Markov chain is assumed to be time-homogeneous?

M3. Line 176: "... true action value may not lie..." Do you mean the "optimal" value function may not lie in the subspace...?

M4. Line 181: `In this case, there are more chances...' Do you have any evidence for this statement?

**Limitations:**

Yes, the authors have discussed the limitations of their work. However, there are more serious issues with the work which I have highlighted.

---

> ### Author Rebuttal · Authors · 2024-08-06
>
> **Q1** *The work studies only the case of a fixed behavior policy. This approach is extremely restrictive and practically not very useful .... Specifically, as stated by Melo, Meyn, and Ribeiro (2008), ..., the behavior policy would need to be close to the optimal policy, ...., the policy estimated by the algorithm could be significantly different from the optimal policy. While Lemma 3.7 provides some guarantees, it is unclear how this result relates ... to the optimal policy. .... , the current work does not address  $\epsilon$-greedy exploration. Under $\epsilon$-greedy exploration, several recent studies ... have shown that Q-learning ... suffers from various issues beyond instability. .... It remains unclear if the regularization term proposed in the present work would effectively address these issues with $\epsilon$-greedy exploration.*
>
> **A1** We thank the reviewer for the insightful comments. In the following, we clarify the concerns raised by the reviewer:
>
> 1) We would like to first note that our assumption of a fixed behavior policy is standard in the literature [4,5]. It is important to note that even under such fixed behavior policy assumption, the convergence of Q-learning with linear function approximation has not been fully explored yet. We proposed a simple and convergent algorithm compared to previous works that guarantee convergence under linear function approximation.
>
> 2) We would like to emphasize that our work does not rely on the assumption used in Melo, Meyn, and Ribeiro (2008). Specifically, we do not require the behavior policy to be close to the optimal policy. In fact, we only require the state-action probability induced by the fixed behavior policy to be non-zero, which is a standard assumption in the literature [4,5].
>
> 3) Lemma 3.7 provides a tight bound on the error of the estimated solution and the true solution, $Q^*$. If the error bound with $Q^*$ is small enough, then the estimated policy will be close to the optimal greedy policy.
>
> As the reviewer suggested, an interesting avenue for future research would be to investigate these properties under an epsilon-greedy behavior policy. Following the reviewer's comments, we have added the discussion in the revised manuscript.
>
> **Q2** *The current work only studies the setting where the state-action sequence is sampled in an IID fashion in each iteration. Do you think the results carry over to the scenario with Markovian samples?*
>
> **A2**  Thank you for the valuable comments. We believe our result can be also extended to the Markovian sample cases. In [6], the authors demonstrated an extension of the Borkar and Meyn Theorem to the Markovian sample cases. Since our proof relies on Borkar and Meyn Theorem, our result can also be extended to establish convergence under Markovian samples. According to the reviewer's comment, we have incorporated this result into the revised manuscript.
>
> **Q3** *Are there any realistic examples of the feature matrix for which the condition in (11) is guaranteed?*
>
> **A3** We thank the reviewer for the constructive comments. If we scale the feature matrix such that $\max(||X||,||X^{\top}||)<1$, then choosing $\eta>2$ satisfies the condition (11) by Lemma 3.3 in the manuscript. Scaling the values of feature matrix is a commonly employed technique in the both theoretical literature or in practice. Consequently, the condition in (11) can be easily met in practice. Following the reviewer's comments, we have added the related discussion in the revised manuscript.
>
> **Q4** *l 90 :  Shouldn't you emphasize what $s_0$'s distribution is?*
>
> **A4** Thank you for pointing out important insights on the paper. In our analysis, we assume that the state-action distribution to satisfy $d(s_0,a_0)>0$ for all $(s_0,a_0)\in{\cal S}\times{\cal A}$. Moreover, if we assume an ergodic Markov chain, the initial distribution does not matter because for any initial distribution, it will converge to stationary distribution of the Markov chain. We have added the discussion in the main manuscript.
>
> **Q5** *l 96 : Which Markov chain is assumed to be time-homogeneous?*
>
> **A5** Thank you for the insightful comments. A Markov chain is said to be time homogeneous if the transition probability does not change over time. We will clarify this in the revision.
>
> **Q6** *l 176 : "... true action value may not lie..." Do you mean the "optimal" value function may not lie in the subspace...?*
>
> **A6** Thank you for the valuable comments. As the reviewer mentioned, it means that optimal action-value function, $Q^*$, may not lie in the subspace. We will clarify this in the revised manuscript.
>
>
> **Q7** *l 181: `In this case, there are more chances...' Do you have any evidence for this statement?*
>
> **A7** Thank you for carefully investigating our manuscript. This is an intuitive result based on the following facts: The Bellman equation $X \theta^* = {\cal T} X \theta^*$ hardly admits a solution because the right-hand side in general does not lie in the column space of $X$ due to the Bellman operator $\cal T$. This situation can be partially alleviated by projecting the right-hand side ${\cal T} X \theta^*$ onto the column space of $X$ using the projection operator $\Gamma$: $X \theta^* = \Gamma {\cal T} X \theta^*$.
> According the reviewer's comment, the corresponding discussions will be modified in the revised manuscript in order to further clarify this intuition.
>
> **References**
>
> [4] Sutton, Richard S., et al. "Fast gradient-descent methods for temporal-difference learning with linear function approximation." Proceedings of the 26th annual international conference on machine learning. 2009.
>
> [5] Lee, Donghwan, and Niao He. "A unified switching system perspective and ODE analysis of Q-learning algorithms." arXiv preprint arXiv:1912.02270 (2019).
>
> [6] Liu, Shuze, Shuhang Chen, and Shangtong Zhang. "The ODE Method for Stochastic Approximation and Reinforcement Learning with Markovian Noise." arXiv preprint arXiv:2401.07844 (2024).

---

> ### Author Response · Authors · 2024-08-13
>
> We appreciate the reviewer's time and effort in reviewing our manuscript. If there are any additional concerns, please let us know since the discussion period is coming to its end. Otherwise, we kindly request a re-evaluation of our manuscript.

---

> > ### Comment · Reviewer_haoJ · 2024-08-13
> >
> > I can never imagine a scenario where a fixed behavior policy will ever be used. However, the bounds given in Lemma 3.7 seem useful and could perhaps lead to some insights that could be exploited in the future for designing more effective algorithms. Hence, I have decided to change my score from 3 to 5.

---

> > > ### Author Response · Authors · 2024-08-14
> > >
> > > We thank the reviewer for the engagement in the discussion period. As the reviewer mentioned, fixed behavior policy is not a practical assumption. However, considering that this is a standard assumption in the literature in proving convergence of Q-learning or TD-learning, we believe our assumption aligns with the existing literature. As the reviewer mentioned, it would be an important direction to further explore on this topic. Following the reviewer's comments, we will incorporate this into the revised manuscript. We thank the reviewer for the time and effort in reviewing our paper.

---

### Author Rebuttal · Authors · 2024-08-06

We sincerely appreciate the reviewers’ constructive comments for our manuscript. The comments are valuable for improving the quality of our paper and provide important guidance for our research.

In the following, we address the concerns commonly raised by the reviewers.

**G1** Reviewers yA4E and x1RW  raised concerns regarding the comparison with existing works that guarantee the convergence of Q-learning with linear function approximation under mild conditions. The algorithms compared include Coupled Q-learning (CQL), GreedyGQ, and algorithms with target-network updates (targetQ).

1) GreedyGQ algorithm is guaranteed to converge to some solution. However, the solution is not a solution of the projected Bellman equation, of which its quality is difficult to quantify. However, we provide a tight bound for our solution in the error bound in Lemma 3.7. Moreover, it is a two-time scale algorithm which is known to be slower than a single-time scale algorithm like ours.


2)  CQL is also a two-time scale algorithm whereas our algorithm is a single-time scale algorithm. As noted previously, two-time scale algorithms are often known to be slower than single-time scale algorithm. This is demonstrated experimentally in Section 6 in the manuscript. Moreover, as can be in the Figure 1c in the PDF file attached in the global response, the algorithm is sensitive to the scaling of the feature matrix, and the solution is not a solution of the projected Bellman equation, of which its quality is difficult to quantify.


3) To guarantee the convergence of targetQ (which uses target network update), it requires a projection or truncation method. This causes additional complexity in its implementation. The resulting solution lacks interpretability, as it may lie on the boundary of the projection or truncation ball. Lastly, the target network update can slow down the convergence rate, which can be verified in Section 6 in the manuscript.

**G2** Reviewers yA4E and cMMt raised concerns regarding the terminology of shrinking and blowing phase in Figure 1(c). We consider the scenario that $|| \gamma\Gamma ||\_{\infty} < 1$ holds, which implies that $|| \gamma \Gamma_0 ||\_\infty < 1$ because $\Gamma = \Gamma_\eta$ when $\eta = 0$. The figure implies that as $\eta\to\infty$, $\gamma \Gamma_\eta$ can potentially move outside of the unit ball, and this phase is indicated with the term **blowing up** phase. However, since $\lim\_{\eta\to\infty } {|| \gamma \Gamma _\eta ||\_\infty } = 0$, we know that $\gamma \Gamma\_\eta$ will eventually converge to the origin and move inside the unit ball. This behavior is indicated by the **shrinking** phase in the figure.

- Additional proofs

**G3**  (Response to Q5 of Reviewers yA4E )

**Lemma** ( Continuity of $\theta^*_{\eta}$ in terms of $\eta$ )
Let $\eta_0$ be a non-negative real valued constant. Suppose $\gamma||\Gamma_{\eta_0}||\_{\infty}<1$. Then, $\theta^*_{\eta}$ is continuous at $\eta_0$.

*Proof)*
Note that $\Gamma_{\eta}$ is continuous function of $\eta$, and we have,
\begin{align*}
    \Gamma_{\eta_0+\eta} = \Gamma_{\eta_0} + O(\eta),
\end{align*}
where $O(\cdot)$ stands for the big O notation. Therefore,
\begin{align*}
    || X\theta^*_{\eta_0+\eta}-X\theta_{\eta_0}^*||\_{\infty} =& || \Gamma_{\eta_0+\eta}\mathcal{T}X\theta^*_{\eta_0+\eta}-\Gamma_{\eta_0} \mathcal{T}X\theta^*_{\eta_0} ||\_{\infty}\\\\
     \leq &  ||\Gamma_{\eta_0} {\cal T} X (\theta^*_{\eta_0+\eta}-\theta^*_{\eta_0}) ||\_{\infty}+O(\eta)\\\\
     \leq & \gamma ||\Gamma_{\eta_0} ||\_{\infty} ||X (\theta^*_{\eta_0+\eta}-\theta_{\eta_0}^*)||\_{\infty}+O(\eta).
\end{align*}

The first equality follows from the definition of $\theta^*_{\eta_0+\eta}$ and $\theta^*_{\eta_0}$. The second inequality follows from triangle inequality. The last inequality follows from the contraction property of the Bellman operator.
Therefore, we have
\begin{align*}
 ||\theta^*_{\eta_0+\eta}-\theta^*_{\eta_0}||\_{\infty} \leq C   ||X\theta^*_{\eta_0+\eta}-X\theta^*_{\eta_0}||_{\infty} \leq O(\eta),
\end{align*}
where the first inequality holds because $X$ is full-column rank matrix, and $C$ is a universal constant. This completes the proof.
*Q.E.D*

We sincerely appreciate the reviewer's feedback. In response to the comments, we have incorporated the relevant discussions into the revised version.

---

### Decision · Program_Chairs · 2024-09-25

**Decision:**

Accept (poster)

**Comment:**

The reviewers of the paper agreed that the contribution is both interesting and very relevant, as Q-learning with function approximation is a widely used algorithm. The reviewers pointed out that the theoretical analysis is the same as in Lee and He 2019, and this was a major reason to reject the paper. The reviewers compared their reviews, discussed the contributions of the paper, and there most critical reviewers were not against acceptance at the end.